# E³-PRUNER: TOWARDS EFFICIENT, ECONOMICAL, AND EFFECTIVE LAYER PRUNING FOR LARGE LANGUAGE MODELS

## ABSTRACT

With the increasing size of large language models, layer pruning has gained increased attention as a hardware-friendly approach for model compression. However, existing layer pruning methods struggle to simultaneously address key practical deployment challenges, including performance degradation, high training costs, and limited acceleration. To overcome these limitations, we propose E³-PRUNER, a task-Effective, training-Economical and inference-Efficient layer pruning framework. E³-PRUNER introduces two key innovations: (1) a differentiable mask optimization method using a Gumbel-TopK sampler, enabling efficient and precise pruning mask search; and (2) an entropy-aware adaptive knowledge distillation strategy that enhances task performance. Extensive experiments over diverse model architectures and benchmarks demonstrate the superiority of our method over state-of-the-art approaches. Notably, E³-PRUNER achieves 96% accuracy, a mere 0.8% drop from the original model (96.8%) on MATH-500 when pruning 25% layers of Qwen3-32B, outperforming existing SOTA (95%), with a $1.33\times$ inference speedup by consuming merely 0.5B tokens (0.5% of the post-training data volume).

## 1 INTRODUCTION

In recent years, guided by scaling laws (Kaplan et al., 2020; Hoffmann et al., 2022), Large Language Models (LLMs) (Touvron et al., 2023; Grattafiori et al., 2024; Yang et al., 2025b;a; DeepSeek-AI et al., 2025b; Team et al., 2025a) have demonstrated substantial advances in performance, achieving unprecedented results across a wide range of tasks and signaling the advent of the artificial intelligence era. However, this performance gain has been accompanied by an exponential growth in model parameters up to trillions of parameters (Team et al., 2025b), which poses increasing challenges for the deployment. To obtain compact LLMs, various model compression methods have been explored, such as pruning (Han et al., 2015; Frantar & Alistarh, 2023; Sun et al., 2024; Chen et al., 2025b), quantization (Frantar et al., 2023; Lin et al., 2024; Xiao et al., 2023; Sun et al., 2025), and knowledge distillation (Hsieh et al., 2023; Hinton et al., 2015).

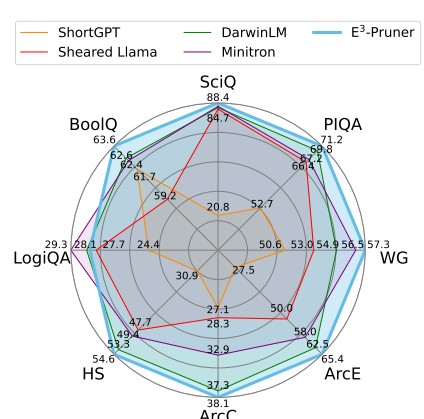

Figure 1: Comparisons between E³-PRUNER and current state-of-the-art structural pruning models in LLaMA-2-7B with 60% pruning ratio.

Among these methods, pruning directly reduces the number of model parameters and effectively shrinks the size of the model. Currently, pruning approaches can be primarily categorized by granularity. Although unstructured pruning (Frantar & Alistarh, 2023; Sun et al., 2024; Frankle & Carbin, 2019) achieves minimal accuracy loss through weight-level sparsity, its irregular patterns fail to deliver practical acceleration benefits. Structured pruning (Ma et al., 2023; Xia et al., 2024; Muralidharan et al., 2024; Tang et al., 2025) operates at a coarse granularity by removing entire structural components, such as attention heads and channels. This approach is generally more amenable to real-world deployment. However, the

irregularity in the resulting pruned architectures still limits achievable speedups. Layer pruning, which eliminates entire transformer blocks, has emerged as a promising alternative capable of delivering scalable inference acceleration. Nevertheless, the significant accuracy degradation often exceeds acceptable levels for practical applications. Moreover, the substantial computational cost associated with mask searching and subsequent fine-tuning undermines its economical viability. These challenges underscore the necessity for more refined pruning strategies that can simultaneously optimize across all key dimensions to deployment.

In this paper, we introduce $E^3$-PRUNER, a task-$\underline{\text{Effective}}$, training-$\underline{\text{Economical}}$ and inference-$\underline{\text{Efficient}}$ layer pruning approach. $E^3$-PRUNER operates in two distinct phases: a *searching stage* and a *fine-tuning stage*. During the *searching stage*, a differentiable Gumbel-TopK sampler substitutes the deterministic top-$k$ operator, enabling gradient-based optimization and facilitating precise and efficient learning of the pruning mask via backpropagation. In the *fine-tuning stage*, an adaptive knowledge distillation strategy is employed, incorporating token-wise re-weighting to emphasize critical token representations and improve knowledge transfer from the teacher model. Extensive experiments across a comprehensive suite of model architectures and benchmarks demonstrate that $E^3$-PRUNER not only achieves superior task performance preservation, but also significantly improves token efficiency. As shown in Figure 1, when pruning 60% of the layers from LLaMA-2-7B (Touvron et al., 2023), $E^3$-PRUNER maintains remarkable higher accuracy. More remarkably, in the challenging scenario of pruning 25% of layers from Qwen3-32B (Yang et al., 2025a), $E^3$-PRUNER is the only approach capable of limiting performance degradation to less than 1% on the MATH-500 (Lightman et al., 2023) while requiring only 0.5% of original post-training data volume, demonstrating its exceptional capability to preserve reasoning abilities.

We summarize our contributions as follows:

- We identify the practical challenges in layer pruning and propose $E^3$-PRUNER, which establishes an efficient, economical and effective paradigm for layer pruning.
- We propose a differentiable Gumbel-TopK sampler that facilitates both efficient and precise mask optimization. This is further enhanced by an adaptive knowledge distillation strategy employing token-wise re-weighting to preserve the performance of the pruned model.
- Through extensive experiments across multiple model architectures and diverse benchmarks, we validate the effectiveness of $E^3$-PRUNER. Our ablation studies and component analyses further provide insights into the mechanisms behind our method's success.

## 2 PRELIMINARIES

### 2.1 LLM PRUNING

**Notations.** We begin with necessary notations for a conventional Transformer architecture, the building block for modern LLMs. We denote the $l$-th Transformer layer as $f(\mathbf{X}_l; \theta_l)$, with $\mathbf{X}_l$ and $\theta_l$ representing its input activations and associated parameters, respectively. Prevalent LLMs adopt the pre-norm architecture, where the input to the $(l+1)$-th layer $\mathbf{X}_{l+1}$ can be obtained by

$$\mathbf{X}_{l+1} = \mathbf{X}_l + f(\mathbf{X}_l, \theta_l). \tag{1}$$

Pruning is a popular solution to obtain lightweight large language models. The key idea of network pruning is to identify and discard redundant parameters. Depending on the categories of discarded parameters, prevalent pruning algorithms can be categorized into unstructured pruning (i.e., removing model parameters) (Frantar & Alistarh, 2023; Sun et al., 2024; Frankle & Carbin, 2019), structured pruning (i.e., removing channels or attention heads) (Ma et al., 2023; Xia et al., 2024; Tang et al., 2025), and layer pruning (Men et al., 2024; Song et al., 2024; Kim et al., 2024; Chen et al., 2025b; Muralidharan et al., 2024). In this work, we focus on layer pruning due to its simplicity as the inference speed-up scales in proportion to the number of pruned layers.

Consider removing the $l$-th transformer layer, the subsequent $(l+1)$-th layer takes the output $f(\mathbf{X}_{l-1}, \theta_{l+1})$ of the $(l-1)$-th layer as its own input, i.e.,

$$\mathbf{X}_{l+1} = \mathbf{X}_{l-1} + f(\mathbf{X}_{l-1}, \theta_{l+1}). \tag{2}$$

**Prior Works.** A key research in layer pruning is to determine which layer to discard. Prior studies have investigated a variety of methods, based on either different pruning metrics (Men et al., 2024;

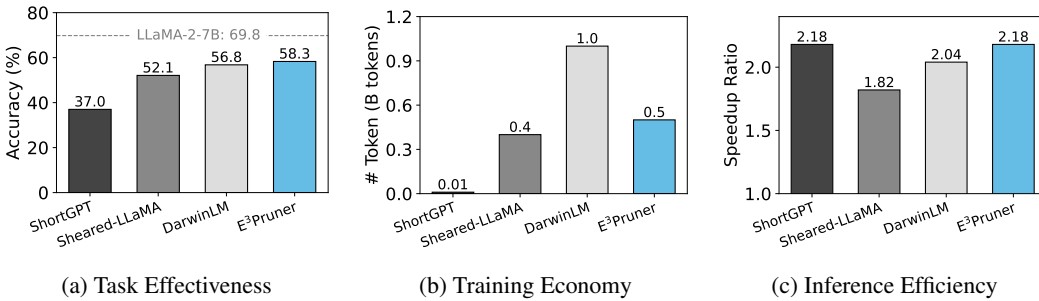

|            (a) Task Effectiveness            |            (b) Training Economy            |            (c) Inference Efficiency            |

Figure 2: The comparisons of task effectiveness, training economy and inference efficiency among existing layer pruning methods. All experiments are done on LLaMA-2-7B model under 60% sparsity.

Chen et al., 2024) or the neural architecture search (Xia et al., 2024; Tang et al., 2025; Zhao et al.). Below we introduce three representative works and more related works can be found in Appendix A:

− **ShortGPT** (Men et al., 2024) is a training-free layer pruning method based on the proposed block importance metric. Specifically, the layer redundancy is measured by the cosine similarity between the activations of different layers on calibration dataset $\mathcal{D}$. The pruned layer indices $l^*$ are thus obtained by repeatedly applying $l^* = \arg\max_l \mathbb{E}_{i \in \mathcal{D}}(\mathbf{X}_{i,l} \cdot \mathbf{X}_{i,l+1}) / (\|\mathbf{X}_{i,l}\| \|\mathbf{X}_{i,l+1}\|)$.

− **DarwinLM** (Tang et al., 2025) is an neural architecture search based pruning method. Aside from layer pruning, DarwinLM also includes the MLP widths and attention heads of LLMs into the search space. For each iteration of the evolution, each candidate undergoes lightweight training for better offspring selection.

− **Sheared-LLaMA** (Xia et al., 2024) employs differentiable pruning via the Gumbel-Softmax reparameterization and sparsity regularization. Each layer is assigned with a learnable soft mask, where the final pruning strategy is obtained by taking the top-$k$ selection of the mask.

## 2.2 PRACTICAL CHALLENGES FOR LAYER PRUNING

The practice of LLM pruning has high demands on inference speed-up without compromising the task accuracy. Moreover, there is usually limited time and computation resource that supports intensive fine-tuning. Based on these three dimensions, we compare the above pruning methods, and an overview is shown in Figure 2.

**Task Effectiveness.** The effectiveness of layer pruning (i.e., the task accuracy) is always of the highest priority in practice. In Figure 2a, we report the mean accuracy across QA tasks in LLaMA-2-7B (Touvron et al., 2023) with 60% sparsity from state-of-the-art methods. As can be observed, ShortGPT (Men et al., 2024) exhibit significantly lower accuracy compared to other approaches, indicating that performance recovery training after pruning remains necessary. Furthermore, despite the fact that all other methods rely on a training process, the performance gap among them underscores the critical importance of obtaining an accurate pruning mask.

**Training Economy.** It is critical to recover the performance of pruned LLM with minimal training cost. For fair and qualitative comparisons, we adopt the training tokens as the metric of training economy. Figure 2b shows that 1) ShortGPT has nearly zero training cost as it is a training-free method; 2) DarwinLM consumes around 1B tokens during the expensive evolution of architecture off-springs; and 3) differentiable search based methods like Sheared-LLaMA and our proposed E$^3$-PRUNER take only 0.4B-0.5B training tokens, demonstrating the potential of training economy.

**Inference Efficiency.** As illustrated in Figure 2c, we evaluate the wall-time speedup ratio on LLaMA-2-7B (Touvron et al., 2023) with 60% sparsity for: uniform structured pruning from Sheared-LLaMA (Xia et al., 2024), non-uniform structured pruning from DarwinLM (Tang et al., 2025), and pure layer pruning in ShortGPT (Men et al., 2024) and our method. Among these, layer pruning achieves the highest acceleration performance.

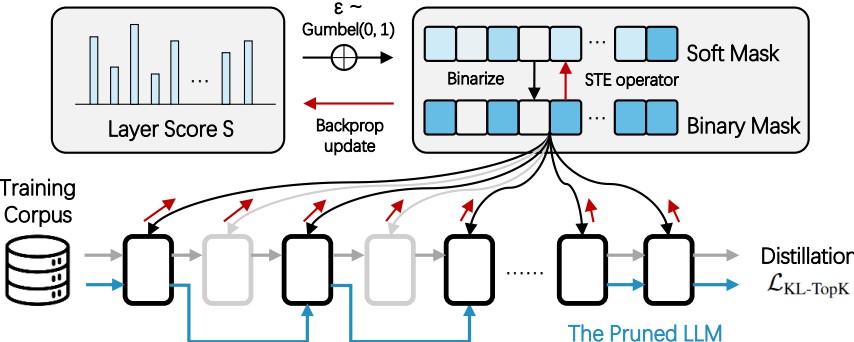

Figure 3: The framework of $E^3$-PRUNER. We employ KL divergence to initialize proposed Gumbel-TopK sampler, which searches for the optimal mask in a differentiable way. Subsequently, the pruned model undergoes efficient adaptive knowledge distillation to restore its performance.

In summary, while current state-of-the-art layer pruning methods exhibit superior advantages in one or two aspects, they can hardly fill all the requirement of task effectiveness, training economy and inference efficiency in the same time.

## 3 METHODS

### 3.1 FORMULATION

In this paper, we introduce $E^3$-PRUNER, a task-Effective, training-Economical and inference-Efficient layer pruning approach for large language models. Inspired by neural architecture search, $E^3$-PRUNER is empowered by an efficient differentiable mask optimization (Section 3.2) in joint effort with an enhanced variant of knowledge distillation (Section 3.3). An overview of $E^3$-PRUNER is presented in Figure 3, and the algorithmic workflow is in Algorithm 1. Specifically, given the $l$-th transformer layer, we assign a pruning mask $m_l \in \{0, 1\}$ for the associated parameter $\theta_l$. The forward computation of the $l$-th layer is thus $\mathbf{X}_{l+1} = \mathbf{X}_l + m_l \cdot f(\mathbf{X}_l, \theta_l)$.

The proposed $E^3$-PRUNER consists of two stages: the *searching stage* and the *fine-tuning stage*. The former aims to identify the pruning masks, while the latter fine-tunes the pruned LLMs for further performance recovery. Specifically, in the *searching stage*, as pruning masks are inherently associated with the LLM parameters, $E^3$-PRUNER optimize both the model parameters $\Theta = \{\theta_1, ..., \theta_L\}$ and the pruning masks $\mathbf{M} = \{m_1, ..., m_L\}$ w.r.t. the training objective $\mathcal{L}$, i.e.,

$$\{\Theta^*, \mathbf{M}^*\} = \arg\min_{\Theta, \mathbf{M}} \mathcal{L}(x; \ \Theta, \mathbf{M}), \quad \text{s.t.} \ \|\mathbf{M}\|_0 = k, \tag{3}$$

where $x \in \mathcal{C}$ denote the training data from corpus $\mathcal{C}$, and $k$ is the number of effective layers. In the *fine-tuning stage*, we keep the pruning mask $\mathbf{M}^*$ and continue to optimize model parameters $\Theta^*$ w.r.t. the objective $\mathcal{L}$. We propose adaptive knowledge distillation as the training objective, i.e., a variant of knowledge distillation that re-weights training tokens, as will be detailed in Section 3.3.

While the formulation in Equation 3 looks promising, the discrete nature of pruning masks $\mathbf{M}$ makes the optimization infeasible. To solve this, prior studies (Xia et al., 2024) adopt Gumbel-Softmax tricks with sparsity regularization (Louizos et al., 2017). However, these methods cannot exactly control the pruning rate, with additional training overhead up to 5 times slower.

### 3.2 DIFFERENTIABLE MASK OPTIMIZATION WITH GUMBEL-TOPK SAMPLER

Instead of directly optimizing the pruning masks $\mathbf{M}$, we introduce layer importance $\mathbf{S} = \{s_1, ..., s_L\}$ as auxiliary variables, where $s_l \in \mathbf{R}$ is the real value indicating the importance of the $l$-th layer. The pruning mask $m_l$ can be thus obtained via the TopK selection, i.e., $m_l = 1$ if $s_l \in \text{TopK}(\mathbf{S}, k)$ and 0 otherwise. However, the discrete operator $\text{TopK}(\mathbf{S}, k)$ is non-differentiable, and we sort to Gumbel-TopK Sampler (Plötz & Roth, 2018) as a continuous relaxation.

**Algorithm 1** $E^3$-PRUNER

**Require:** LLM $\Theta$, total iteration $T$, mask searching iteration $T_M$, layer score $\mathbf{S}$, total layers $L$.
1: **for** $t = 1$ to $T$ **do**
2:     $\tau = 1 - \beta \cdot t/T$, $k' = L - \lceil (L-k) \cdot t/T \rceil$
3:     **if** $t \leq T_M$:
4:         $\mathbf{M} = \text{Gumbel-TopK}(\mathbf{S}, \tau, k')$
5:         $\mathbf{S} = \mathbf{S} - \lambda \cdot \partial \mathcal{L}(\mathbf{x}, \Theta, \mathbf{M})/\partial \mathbf{S}$
6:     $\Theta = \Theta - \lambda \cdot \partial \mathcal{L}(\mathbf{x}, \Theta, \mathbf{M})/\partial \Theta$
7: **end for**
8: **return** Pruned model $\Theta^*$

**Algorithm 2** Gumbel-TopK

**Require:** Layer score $\mathbf{S}$, temperature $\tau$, number of retained layers $k$.
1: $\mathbf{S}_1 = \mathbf{S} + \text{Gumbel}(0,1)$
2: $\tilde{\mathbf{M}}_1 = \mathbf{0}$
3: **for** $i = 1$ to $k$ **do**
4:     $\mathbf{S}_{i+1} = \mathbf{S}_i + \log(\mathbf{1} - \text{Softmax}(\frac{\mathbf{S}_i}{\tau}))$
5:     $\tilde{\mathbf{M}}_{i+1} = \tilde{\mathbf{M}}_i + \text{Softmax}(\frac{\mathbf{S}_i}{\tau})$
6: **end for**
7: $\mathbf{M} = \text{STE}(\text{TopK}(\tilde{\mathbf{M}}_{k+1}, k))$
8: **return** Pruning mask $\mathbf{M}$

**Gumbel-TopK Sampler.** The TopK$(\mathbf{S}, k)$ operator can be equivalently translated to repeatedly picking $k$ largest elements from $\mathbf{S}$. In particular, the Gumbel-Softmax trick first sample Gumbel noise $g \sim \text{Gumbel}(0,1)$, and add it over $\mathbf{S}$, i.e., $\mathbf{S}_1 = \mathbf{S} + g$. Then, for the $i$-th iteration ($i \geq 1$),

$$\tilde{\mathbf{M}}_{i+1} = \tilde{\mathbf{M}}_i + \text{Softmax}(\frac{\mathbf{S}_i}{\tau}), \quad \mathbf{S}_{i+1} = \mathbf{S}_i + \log(1 - \text{Softmax}(\frac{\mathbf{S}_i}{\tau})), \tag{4}$$

where $\tilde{\mathbf{M}}_i$ is the soft $i$-hot mask variable, $\text{Softmax}(\cdot/\tau)$ is the softmax function with temperature $\tau$. Plötz & Roth (2018) has shown that as $\tau \to 0$, $\tilde{\mathbf{M}}_k \to \text{TopK}(\mathbf{S}, k)$. In practice, we initially choose a moderate temperature, and linearly anneal it with the training iterations $t$ as $\tau = 1 - \beta \cdot (t/T)$, where $\beta$ is a coefficient. To ensure pruning in the forward pass, we discretize the soft mask $\tilde{\mathbf{M}}_k$ as $\mathbf{M}$, and adopt the straight-through estimator (STE) (Bengio et al., 2013) for the backward pass to ensure the differentiability. The whole workflow of Gumbel-TopK sampler can be found in Algorithm 2. Notably, all operations—noise injection, softmax, and mask updates—introduce negligible computational overhead and are executed only once per forward pass, thus preserving training economy.

**Initialization of Layer Importance.** A good initialization of layer importance scores $\mathbf{S}$ could effectively reduce the training cost and task performance. For the $l$-th layer, we initialize $s_l$ as the Kullback-Leibler (KL) divergence $\mathcal{D}_{\text{KL}}(\cdot, \cdot)$ between the pruned model and the original model, i.e., $\mathbb{E}_{x \sim \mathcal{C}} [D_{\text{KL}}(f(\mathbf{x}; \Theta_{-l}) \| f(\mathbf{x}; \Theta))]$, where $\Theta_{-l}$ denotes the LLM parameters without the $l$-th layer.

**Progressive Layer Pruning.** To ensure a smooth transition between the original model and the pruned LLM, we employ a curriculum schedule that gradually increases the pruning ratio, which is well-established for preserving model performance (Frankle & Carbin, 2019; Zhu & Gupta, 2017). We linearly increase the number of pruned layers as the training iterations increase, until reaching the target retained layers $k$, i.e., $k' = L - \lceil (L-k) \cdot t/T \rceil$.

### 3.3 ADAPTIVE KNOWLEDGE DISTILLATION

Now we are ready to introduce adaptive knowledge distillation, a compelling training objective to enhance the task effectiveness of $E^3$-PRUNER. The key idea behind this technique is to transfer the knowledge from the original LLM to the pruned model, where each token is re-weighted adaptively based on its entropy. A practical challenge to implement knowledge distillation is the computation overhead, i.e., it is rather expensive to compute the logits of both teacher and student models on the fly. Therefore, we pre-compute and save the top-$K$ logits of the teacher model for each token. During training, these logits are loaded back to compute the KL divergence with the pruned model, i.e.,

$$\mathcal{L}_{\text{token}} = \sum_{c \in \mathcal{S}_i^K} \text{Softmax}(\mathbf{z}_t^{(i)})_c \cdot \log \left( \frac{\text{Softmax}(\mathbf{z}_t^{(i)})_c}{\text{Softmax}(\mathbf{z}_s^{(i)})_c} \right), \quad \mathcal{L}_{\text{KL-TopK}} = \frac{1}{N} \sum_{i=1}^{N} \mathcal{L}_{\text{token}}, \tag{5}$$

where $N$ is the total number of tokens, $\mathbf{z}_t^{(i)}$ and $\mathbf{z}_s^{(i)}$ denote the pre-softmax logits of the teacher and student models for the $i$-th token, $\mathcal{S}_i^K$ contains the indices of the top-$K$ values in the teacher's distribution. This approach introduces minor storage requirements while avoiding the computational cost of loading the full teacher model, making the training more economical.

Based on Equation 5, we argue that different tokens contribute unequally to model performance and should be treated accordingly. Existing works Wang et al. (2025) show that token entropy $\mathcal{H}$ serves as an effective metric to identify these informative tokens. This finally leads to the objective of adaptive knowledge distillation in Equation 3, i.e.,

$$\mathcal{L} = \frac{1}{N} \sum_{i=1}^{N} \mathcal{H}(\text{Softmax}(\mathbf{z}_t^{(i)})_c) \cdot \mathcal{L}_{\text{token}}. \tag{6}$$

This formulation assigns greater weight to high-entropy tokens, which often correspond to uncertain or critical reasoning steps (Wang et al., 2025; Chen et al., 2025a).

## 4 EXPERIMENTS

### 4.1 SETUP

**Models and Datasets.** To comprehensively validate the efficacy of $\mathrm{E}^3$-PRUNER, we conduct extensive experiments across multiple LLM architectures: LLaMA-2-7B (Touvron et al., 2023), Qwen2.5-14B-Instruct (Yang et al., 2025b), Qwen3-32B (Yang et al., 2025a), and DeepSeek-R1 (DeepSeek-AI et al., 2025a). This selection covers diverse model scales and training stages (from pre-training to post-training), thereby ensuring a thorough evaluation of the method's generalizability. For the training corpus, in line with prior works, we utilize the open-source Fineweb-Edu (Lozhkov et al., 2024) dataset for LLaMA-2-7B and Qwen2.5-14B-Instruct. For the reasoning-oriented models Qwen3-32B and DeepSeek-R1, we use the AM-DeepSeek-R1-Distilled-1.4M dataset (Zhao et al., 2025), which provides more reasoning content in the text.

**Baselines.** We perform systematic comparisons between $\mathrm{E}^3$-PRUNER and state-of-the-art pruning approaches: ShortGPT (Men et al., 2024), Sheared-LLaMA (Xia et al., 2024), DarwinLM (Tang et al., 2025), and Minitron (Muralidharan et al., 2024). Where available, official model checkpoints are used to reproduce reported results. Otherwise, each baseline is carefully re-implemented to ensure fair comparison. For Sheared-LLaMA, we adhere to the original implementation while limiting adaptations to layer pruning mask learning. For Minitron, we follow the published procedure by first computing BI scores (Men et al., 2024) to identify redundant layers, then performing knowledge distillation-based recovery. To ensure fair comparisons among the baselines, we keep the identical training data, fixed token budgets, a unified distillation framework, and consistent pruning ratios. We leave more pruning specifications and training hyperparameters in Appendix D.

**Evaluation Details.** For LLaMA-2-7B and Qwen2.5-14B-Instruct, we follow established evaluation protocols by conducting zero-shot assessments on SciQ (Welbl et al., 2017), PIQA (Bisk et al., 2020), WinoGrande (Sakaguchi et al., 2021), ARC-easy (Clark et al., 2018), LogiQA (Liu et al., 2020) and BoolQ (Clark et al., 2019), along with few-shot evaluations on HellaSwag (Zellers et al., 2019) (10-shot), ARC Challenge (Clark et al., 2018) (25-shot) and MMLU (Hendrycks et al., 2021) (5-shot), utilizing the standardized lm-evaluation-harness framework (Gao et al., 2024) [1] to ensure consistency. For reasoning models, we evaluate models on complex reasoning tasks including MATH-500 (Lightman et al., 2023), AIME-2024, AIME-2025, GPQA-Diamond (Rein et al., 2023), LiveCodeBench (Jain et al., 2024) and a representative 2000-question subset of MMLU-Pro (Wang et al., 2024) for efficient yet comprehensive assessment. Given the limited size of AIME benchmarks, we conduct 8 independent evaluation runs and report mean accuracy to reduce measurement variance.

### 4.2 MAIN RESULTS

**Results on LLaMA-2-7B.** As shown in Table 1, $\mathrm{E}^3$-PRUNER achieves superior performance on most downstream tasks after removing 60% of the layers of LLaMA-2-7B, attaining an average accuracy of 58.3%—surpassing both Minitron (55.3%) and DarwinLM (56.8%). Although Sheared-LLaMA also employs learnable masks, $\mathrm{E}^3$-PRUNER exhibits significantly better results, attributable to its unique equivalence-preserving property between mask search and pruning. Moreover, $\mathrm{E}^3$-PRUNER delivers a highest speedup of 2.18× with a modest token budget of 0.5B, highlighting its efficiency inference and economy in training.

---

[1] https://github.com/EleutherAI/lm-evaluation-harness/releases/tag/v0.4.8

Table 1: Results on LLaMA-2-7B and Qwen2.5-14B-Instruct. $E^3$-PRUNER achieves best in performance and speedup. We omit the MMLU results for LLaMA-2-7B as they are close to random for all methods. *: We use the officially released checkpoints to reproduce the reported results.

| Method | Param. | SciQ | PIQA | WG | ArcE | ArcC(25) | HS(10) | LogiQA | BoolQ | MMLU(5) | Avg↑ | Speedup↑ | # Token↓ |
|---|---|---|---|---|---|---|---|---|---|---|---|---|---|
| | | | | | | *LLaMA-2-7B* | | | | | | | |
| Dense | 6.7B | 94.0 | 78.1 | 69.0 | 76.3 | 54.2 | 78.7 | 30.4 | 77.7 | 45.9 | 69.8 | 1.0× | |
| ShortGPT* | 2.7B | 20.8 | 52.7 | 50.6 | 27.5 | 27.1 | 30.9 | 24.4 | 61.7 | - | 37.0 | 2.18× | 0.01B |
| Sheared-LLaMA* | 2.7B | 84.7 | 66.4 | 53.0 | 50.0 | 28.3 | 47.7 | 27.7 | 59.2 | - | 52.1 | 1.82× | 0.4B |
| DarwinLM* | 2.7B | 85.6 | 69.8 | 54.9 | 62.5 | 37.3 | 53.3 | 28.3 | 62.6 | - | 56.8 | 2.04× | 1.0B |
| Minitron | 2.7B | 86.4 | 67.2 | 56.5 | 58.0 | 32.9 | 49.4 | **29.3** | 62.4 | - | 55.3 | 2.18× | 0.5B |
| $E^3$-PRUNER | 2.7B | **88.4** | **71.2** | **57.3** | **65.4** | **38.1** | **54.6** | 28.1 | **63.6** | - | **58.3** | 2.18× | 0.5B |
| | | | | | | *Qwen2.5-14B-Instruct* | | | | | | | |
| Dense | 14.8B | 96.8 | 81.8 | 75.7 | 85.7 | 72.4 | 85.1 | 39.3 | 87.9 | 79.8 | 78.3 | 1.0× | |
| ShortGPT | 8.2B | 31.6 | 56.7 | 50.3 | 30.5 | 27.6 | 34.0 | 24.9 | 53.8 | 24.6 | 37.1 | 1.63× | 0.01B |
| Sheared-LLaMA | 8.2B | 93.4 | 74.8 | 60.4 | 71.3 | 44.4 | 64.1 | 26.0 | 65.9 | 32.1 | 59.2 | 1.63× | 0.5B |
| DarwinLM* | 8.4B | 84.3 | 73.8 | 59.0 | 75.8 | **49.1** | 53.6 | 28.9 | 67.2 | **42.8** | 59.4 | 1.43× | 1.0B |
| Minitron | 8.2B | 91.9 | 73.5 | 62.0 | 72.1 | 44.9 | 64.5 | 28.4 | **70.4** | 32.8 | 60.1 | 1.63× | 0.5B |
| $E^3$-PRUNER | 8.2B | **93.7** | **76.9** | **63.0** | **76.0** | 47.9 | **67.2** | **30.0** | 66.5 | 36.5 | **61.9** | 1.63× | 0.5B |

Table 2: Results on Qwen3-32B and DeepSeek-R1. †: We test the results based on LiveCodeBench-v5, with questions spanning from 2024.08 to 2025.01.

| Method | Param. | MATH-500 | AIME'24 | AIME'25 | GPQA-Diamond | LiveCode-Bench† | MMLU-Pro | ArenaHard | Avg↑ | Speedup↑ | # Token↓ |
|---|---|---|---|---|---|---|---|---|---|---|---|
| | | | | | *Qwen3-32B* | | | | | | |
| Dense | 32.8B | 96.8 | 79.6 | 71.3 | 64.7 | 66.9 | 76.5 | 93.5 | 78.5 | 1.0× | |
| Sheared-LLaMA | 28.9B | 93.2 | 60.0 | 55.4 | 52.0 | 56.3 | 65.1 | 78.6 | 65.8 | | |
| Minitron | 28.9B | 95.4 | 72.1 | 64.2 | **61.6** | 62.9 | **74.8** | **92.3** | 74.8 | 1.13× | 0.5B |
| $E^3$-PRUNER | 28.9B | **96.2** | **76.3** | **65.8** | 61.1 | **65.1** | 74.5 | 91.4 | **75.8** | | |
| Sheared-LLaMA | 25.0B | 92.4 | 58.8 | 49.6 | 40.9 | 36.0 | 55.8 | 67.4 | 57.3 | | |
| Minitron | 25.0B | 95.0 | 68.3 | 61.7 | 59.1 | 57.7 | **70.5** | 85.0 | 71.0 | 1.33× | 0.5B |
| $E^3$-PRUNER | 25.0B | **96.0** | **71.7** | **64.2** | **60.1** | **60.3** | 70.1 | **87.3** | **72.8** | | |
| | | | | | *DeepSeek-R1* | | | | | | |
| Dense | 671B | 97.2 | 77.9 | 67.1 | 73.7 | 65.4 | 82.8 | 96.3 | 80.1 | 1.0× | |
| Sheared-LLaMA | 601B | 96.6 | 71.7 | 57.5 | 66.7 | 55.5 | 80.0 | 93.0 | 74.4 | | |
| Minitron | 601B | **97.0** | 75.0 | **62.9** | 71.2 | 65.1 | **81.8** | 95.0 | 78.3 | 1.13× | 0.5B |
| $E^3$-PRUNER | 601B | **97.0** | **76.3** | 60.8 | **73.2** | **67.3** | 81.2 | **96.2** | **78.9** | | |
| Sheared-LLaMA | 509B | 92.6 | 59.6 | 42.1 | 61.1 | 49.3 | 74.4 | 84.1 | 66.2 | | |
| Minitron | 509B | **95.6** | **68.8** | 55.4 | 69.2 | 59.2 | **80.2** | 89.8 | 74.0 | 1.33× | 0.5B |
| $E^3$-PRUNER | 509B | 95.2 | 65.8 | **60.0** | 69.7 | 59.9 | 79.8 | **90.0** | **74.3** | | |

**Results on Qwen2.5-14B-Instruct.** For Qwen2.5-14B-Instruct, we prune 24 of 48 layers and report the accuracy on downstream tasks in Table 1. Similarly, with the same prune ratio, $E^3$-PRUNER achieves the highest average accuracy of 61.9%, outperforming all baseline methods. Notably, it sets a new state-of-the-art on 6 out of 9 evaluated tasks, demonstrating the efficacy of our approach on instruction-tuned models.

**Results on Qwen3-32B.** To evaluate $E^3$-PRUNER on reasoning-focused models, we conduct experiments with Qwen3-32B, as summarized in Table 2. When 1/8 of the layers are removed, $E^3$-PRUNER achieves an average accuracy of 75.8%, surpassing Minitron (74.8%) and Sheared-LLaMA (65.8%). This advantage persists under more aggressive pruning (1/4 layers removed), where our method maintains an accuracy of 72.8%, compared to 71.0% for Minitron and 57.3% for Sheared-LLaMA. Notably, on MATH-500, $E^3$-PRUNER exhibits a degradation of less than 1% even at the 1/4 pruning ratio, underscoring its robustness and practical viability for scenarios demanding both inference efficiency and mathematical reasoning capability.

**Results on DeepSeek-R1.** We further evaluate the scalability of $E^3$-PRUNER on DeepSeek-R1, a 671B mixture-of-experts model. As shown in Table 2, $E^3$-PRUNER outperforms baselines in

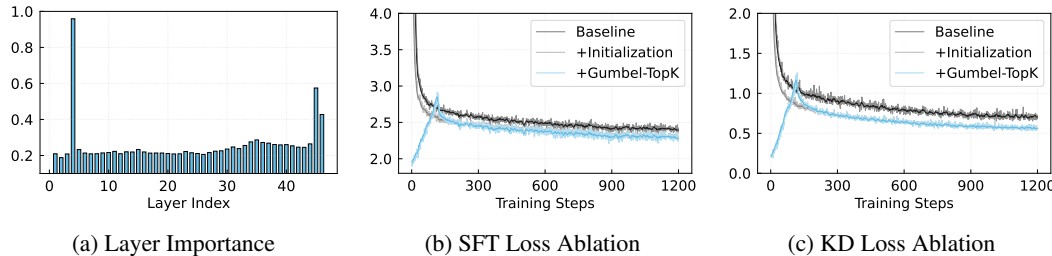

Figure 4: Ablations on Qwen2.5-14B-Instruct (a): Our initialization identifies important layers at the beginning. (b): Proposed components contribute to lower loss in SFT training. (c): Proposed components contribute to lower loss in KD training.

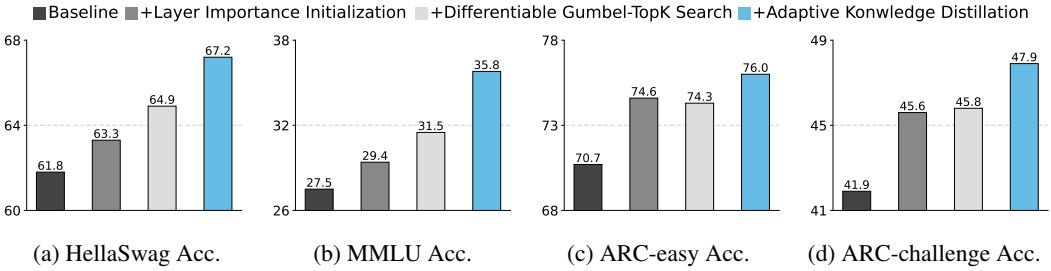

Figure 5: Performance improvement breakdown. By applying proposed $E^3$-PRUNER, we achieve significant performance improvements across diverse benchmarks on Qwen2.5-14B-Instruct.

both 8-layer and 16-layer pruning configurations. When pruning 8 layers, our approach achieves an overall accuracy of 78.9%, exceeding Minitron (78.3%) and Sheared-LLaMA (74.4%), while maintaining near-original performance on most tasks. Under more aggressive pruning with 16 layers removed, the resulting model retains an accuracy of 74.3%, demonstrating the robustness of our method in large-scale model compression with only marginal performance degradation. These results collectively validate the efficacy of $E^3$-PRUNER in effectively compressing large-scale MoE models.

## 4.3 ANALYSIS

**Ablation Study.** We ablate our method's components to assess their individual contributions. Figure 4a shows that our initialization effectively distinguishes important layers for Qwen2.5-14B-Instruct in the initial stage (excluding the first and last layers, which are not pruned). The training curves in Figures 4b and 4c indicate that both the proposed initialization and the Gumbel-Topk search contribute to superior pruning mask, leading to lower losses during either supervised fine-tuning or knowledge distillation. The detailed analysis of the Gumbel-TopK sampler dynamics can be found in Appendix C. Furthermore, Figure 5 provides a detailed breakdown of the performance gains across multiple benchmarks, confirming the effectiveness of each component.

**The Balance between Searching and Fine-tuning.** We further examine the influence of the searching budget, the fraction of total training steps allocated to the *searching stage*, on the performance of our method. As shown in Table 3, $E^3$-PRUNER consistently outperforms existing state-of-the-art approaches across various budget configurations. This indicates that our method exhibits strong robustness with respect to the searching budget. In all other experiments in this paper, a searching budget of one-tenth of the total training steps is adopted.

**Consistency Comparison between SFT and KD.** We demonstrate that knowledge distillation not only achieves substantially higher performance than supervised fine-tuning but also maintains significantly greater behavioral consistency between the pruned and original models. To assess behavioral consistency, we constructed an evaluation set by randomly sampling 100 test cases from each of four domains: MMLU (Hendrycks et al., 2021), GSM8K (Cobbe et al., 2021), IFeval (Zhou et al., 2023), and Humaneval (Chen et al., 2021). The average accuracy on this set is reported

Table 3: Ablation of searching budget. $E^3$-PRUNER consistently outperform state-of-the-art methods across various mask search budget on Qwen2.5-14B-Instruct, demonstrating its robustness.

| Searching Budget | SciQ | PIQA | WG | ArcE | ArcC(25) | HS(10) | LogiQA | BoolQ | MMLU(5) | Avg ↑ |
|---|---|---|---|---|---|---|---|---|---|---|
| 0.05 | 92.0 | 76.6 | 62.8 | 75.6 | 48.6 | 67.0 | 28.0 | 66.5 | 29.0 | 60.7 |
| 0.1 | **93.7** | **76.9** | **63.0** | **76.0** | 47.9 | 67.2 | **30.0** | 66.5 | 35.8 | **61.9** |
| 0.2 | 91.9 | 75.8 | 59.9 | 75.9 | **49.7** | 67.8 | 28.6 | 66.4 | **37.1** | 61.5 |
| 0.5 | 91.0 | 75.9 | 59.7 | 73.7 | 48.3 | 65.3 | 28.3 | **67.0** | 34.3 | 60.4 |

Table 4: Ablation of adaptive KD. Compared to normal KD, adaptive KD effectively improve the performance on Qwen2.5-14B-Instruct.

| KD Type | SciQ | PIQA | WG | ArcE | ArcC(25) | HS(10) | LogiQA | BoolQ | MMLU(5) | Avg ↑ |
|---|---|---|---|---|---|---|---|---|---|---|
| Offline KD | **94.0** | 76.2 | 62.5 | 75.2 | **48.0** | 65.6 | 29.0 | 63.8 | 32.8 | 60.8 |
| Adaptive KD | 93.7 | **76.9** | **63.0** | **76.0** | 47.9 | **67.2** | **30.0** | **66.5** | **35.8** | **61.9** |

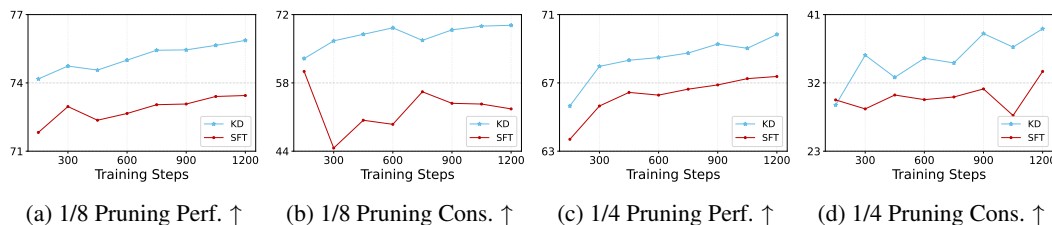

(a) 1/8 Pruning Perf. ↑    (b) 1/8 Pruning Cons. ↑    (c) 1/4 Pruning Perf. ↑    (d) 1/4 Pruning Cons. ↑

Figure 6: Performance and consistency during training. Despite both improving performance, KD effectively improves the consistency while SFT deteriorates consistency in some case.

as the consistency score. Using the Qwen2.5-14B-Instruct model, we conducted comprehensive comparative experiments under varying pruning ratios. As illustrated in Figure 6, both SFT and KD exhibit improved performance of the pruned models as training progresses, indicating a gradual recovery of capabilities. However, the behavioral consistency of SFT-trained models fails to improve effectively with additional steps; indeed, under the 1/8 pruning ratio, it further declines. This suggests that SFT leads to overfitting during recovery, causing the pruned model to diverge from the original model's behavior. In contrast, KD ensures a steady increase in consistency throughout training, effectively preserving behavioral alignment with the original model.

**Effect of Adaptive Knowledge Distillation.** To demonstrate the superiority of our proposed adaptive KD loss over the conventional KD loss, we conduct a comparative experiment on Qwen2.5-14B-Instruct. We prune 50% layers and keep all other hyperparameters unchanged. Results in Table 4 show that while $E^3$-PRUNER with conventional KD loss already achieves competitive performance, adopting the adaptive KD loss further improves the accuracy from 60.8% to 61.9%, demonstrating its effectiveness in identifying and emphasizing the critical tokens that contribute more significantly.

Due to limited space, we defer the experiments on additional model architectures to Appendix B.

## 5 CONCLUSION

In this paper, we first review existing pruning methods and observe that they struggle to simultaneously fulfill three critical requirements for practical deployment: task effectiveness, training economy, and inference efficiency. To overcome this limitation, we propose $E^3$-PRUNER, an effective, economical, and efficient layer pruning approach. $E^3$-PRUNER incorporates a differentiable Gumbel-TopK sampling mechanism to enable efficient and precise optimization of layer masks, along with an adaptive knowledge distillation strategy to improve performance and maintain consistency. Extensive experiments conducted on diverse model architectures and benchmarks demonstrate the superiority of our approach. The results not only confirm the efficacy of our method, but also show that it achieves the best trade-off to date across the challenges for practical deployment.

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

## A  RELATED WORK

**Structured Pruning.**  Structured pruning methods aim to reduce model size by removing entire substructures—such as attention heads, MLP channels, or hidden dimensions—directly from the network. Prior research has proposed various criteria for identifying the least important components. For instance, LLM-Pruner (Ma et al., 2023) identifies coupled structural groups and evaluates their collective importance to guide pruning decisions. SliceGPT (Ashkboos et al., 2024) applies principal component analysis to weight matrices and removes the least significant components, thereby reducing the dimensionality of the weight matrices. Although these methods are computationally lightweight, they often lead to non-negligible performance degradation. More recent approaches, such as Sheared-LLaMA (Xia et al., 2024), achieve state-of-the-art results by incorporating differentiable mask search and dynamic batching during retraining to maintain performance across diverse tasks. Meanwhile, DarwinLM (Tang et al., 2025) and Minitron (Muralidharan et al., 2024) adopt evolutionary search strategies to identify optimal pruning masks, followed by recovery training. However, these techniques typically require iterative training and evaluation of candidate architectures, which incur high computational costs and hinder practical deployment.

**Layer Pruning.**  Layer pruning has recently gained attention as an effective strategy for compressing large language models. In contrast to width pruning, which may result in irregular and hardware-unfriendly structures, layer pruning removes entire Transformer layers—including both attention and feed-forward modules—thereby preserving the regularity of the model and facilitating efficient deployment. Recent studies, including ShortGPT (Men et al., 2024), SLEB (Song et al., 2024), Shortened LLaMA (Kim et al., 2024), and LinearPatch (Chen et al., 2025b), have demonstrated the viability of layer pruning in significantly reducing model depth while maintaining competitive performance. Despite achieving notable inference speedups, existing calibration-based layer pruning methods still struggle to fully preserve the original model's capabilities, which remains a major obstacle to their widespread adoption in deployment.

**Neural Architecture Search.**  Neural Architecture Search (NAS) automates the design of network architectures by searching for optimal sub-networks within a predefined super-net. Typical NAS frameworks focus on optimizing sampling strategies to identify high-performing and efficient architectures. For example, Flextron (Cai et al., 2024) introduces an integrated super-net that encapsulates multiple sub-networks, enabling flexible and adaptive deployment. Similarly, Jet-Nemotron (Gu et al., 2025) utilizes a PostNAS pipeline to design efficient linear attention blocks. These NAS-based approaches primarily target efficient model design and are orthogonal to our work, which focuses on developing a effective, economical, and efficient pruning strategy tailored for real-world deployment scenarios.

## B  ADDITIONAL RESULTS

### B.1  OPENPANGU MODEL RESULTS

We evaluate $E^3$-PRUNER on recently released openPangu-Ultra-MoE-718B (Ascend Tribe, 2025), a large-scale mixture-of-experts language model that incorporates depth-scaled Sandwich-Norm and EP-group load balancing. During both the mask search and recovery phases, we adopt a LoRA fine-tuning paradigm with a rank of 64 applied to all linear projection weights. The model is trained on 1.5 billion tokens sampled from the AM-DeepSeek-R1-Distilled-1.4M corpus (Zhao et al., 2025). Training is conducted over 6,000 iterations using a cosine learning rate scheduler that decays from 1e-4 to 1e-5. For $E^3$-PRUNER, we prune a total of 8 layers, comprising 6 MoE layers and 2 dense layers. In the case of Minitron (Muralidharan et al., 2024), only 6 MoE layers are pruned, as the dense layers exhibit higher importance scores. The results are summarized in Table 5. Although $E^3$-PRUNER prunes two additional layers compared to Minitron, it achieves significantly better performance, with an average performance degradation of only 1.4%, in contrast to the 5.7% performance loss observed with Minitron. These findings further validate the superiority of the proposed $E^3$-PRUNER.

Table 5: Results on openPangu-Ultra-MoE-718B. [†]: We test the results based on LiveCodeBench-v5, with questions spanning from 2024.08 to 2025.01.

| Method | Param. | MATH-500 | AIME'24 | AIME'25 | GPQA-Diamond | LiveCode-Bench[†] | MMLU-Pro | ArenaHard | Avg ↑ | Speedup ↑ | # Token ↓ |
|---|---|---|---|---|---|---|---|---|---|---|---|
| Dense | 718B | 97.8 | 82.5 | 76.7 | 80.8 | 69.5 | 80.2 | 97.5 | 83.6 | 1.0× | |
| Minitron | 644B | 96.6 | 74.6 | 64.6 | 73.2 | 62.5 | 80.1 | 93.7 | 77.9 | 1.13× | 1.5B |
| E³-PRUNER | 641B | **96.8** | **81.3** | **72.9** | **78.8** | **68.8** | **80.5** | **96.2** | **82.2** | 1.13× | 1.5B |

Table 6: DeepSeek-R1 speedup results. Our 8-layer pruned model delivers a consistent 1.13× speedup across diverse experimental settings, demonstrating the practical deployability of E³-PRUNER.

| Batchsize | Original TTFT (ms) ↓ | Pruned TTFT (ms) ↓ | Prefill Speedup ↑ | Original TPOT (ms) ↓ | Pruned TPOT (ms) ↓ | Decode Speedup ↑ |
|---|---|---|---|---|---|---|
| 1 | 7014.00 | 6187.05 | 1.13 | 70.63 | 62.03 | 1.14 |
| 2 | 10487.66 | 9230.56 | 1.14 | 69.65 | 61.98 | 1.12 |
| 8 | 12463.26 | 10857.67 | 1.15 | 70.97 | 62.93 | 1.13 |
| 64 | 13156.46 | 11451.40 | 1.15 | 73.19 | 63.83 | 1.15 |
| 128 | 18973.38 | 16459.24 | 1.15 | 74.39 | 66.16 | 1.12 |

## B.2 DEEPSEEK-R1 SPEEDUP RESULTS

We evaluate the speedup ratio achieved by our layer-pruned DeepSeek-R1 model using the vLLM (Kwon et al., 2023) framework in a realistic 64-card deployment environment. As summarized in Table 6, under the quantization of w8a8 and with input and output lengths configured to 2K and 1K tokens, respectively, the 8-layer pruned model consistently demonstrates a 1.13× speedup in both Time to First Token (TTFT) and Time per Output Token (TPOT) across a range of batch sizes. These results affirm the practical viability and deployment efficiency of the proposed method.

## C GUMBEL SCORES OPTIMIZATION TRAJECTORY

Figure 7 illustrates the optimization trajectory of the Gumbel scores during the mask search phase. In Figure 7a, we present the evolution of layer-wise scores throughout the search process. It can be seen that as the search progresses, certain layers initially selected for retention exhibit a decrease in scores and are eventually pruned. In contrast, other layers that demonstrate greater importance are identified and retained in later stages of the search. Figure 7b displays the layers retained during the forward pass, sampled by the proposed Gumbel-TopK sampler. Notably, due to our gradual strategy, the number of retained layers decreases progressively. In the early phases, the Gumbel-TopK sampler efficiently explores the mask search space, allowing nearly every layer to be potentially pruned. In later stages, as the scores converge and the sampling temperature decreases, the mask selection stabilizes, indicating that an optimal mask has been identified.

## D IMPLEMENTATION DETAILS

We employs distinct pruning strategies that are tailored to different model architectures. For LLaMA-2-7B and Qwen2.5-14B-Instruct, we adopt target pruning ratios of 60% and 50%, respectively, consistent with established practices in prior research. For the reasoning models Qwen3-32B and DeepSeek-R1, we implement layer removal configurations of 12.5% / 25% layers based on empirical performance thresholds. The pruning process begins with a layer importance estimation using 40 calibration samples, followed by a Gumbel-TopK mask searching and subsequent recovery training using 0.5B tokens. To address memory constraints, we employ LoRA (Hu et al., 2021) fine-tuning for DeepSeek-R1 pruning. The gradual mask search occupies the first 10% of the training budget. For the remaining stages, we switch to the adaptive knowledge distillation to preserve task effectiveness. In a 64 NPUs setting, the pruning for LLaMA-2-7B can be done in 0.5 hours, while it take about 1.5 hours for Qwen2.5-14B-Instruct. Complete specifications of pruning configurations and training hyperparameters are provided in Table 7.

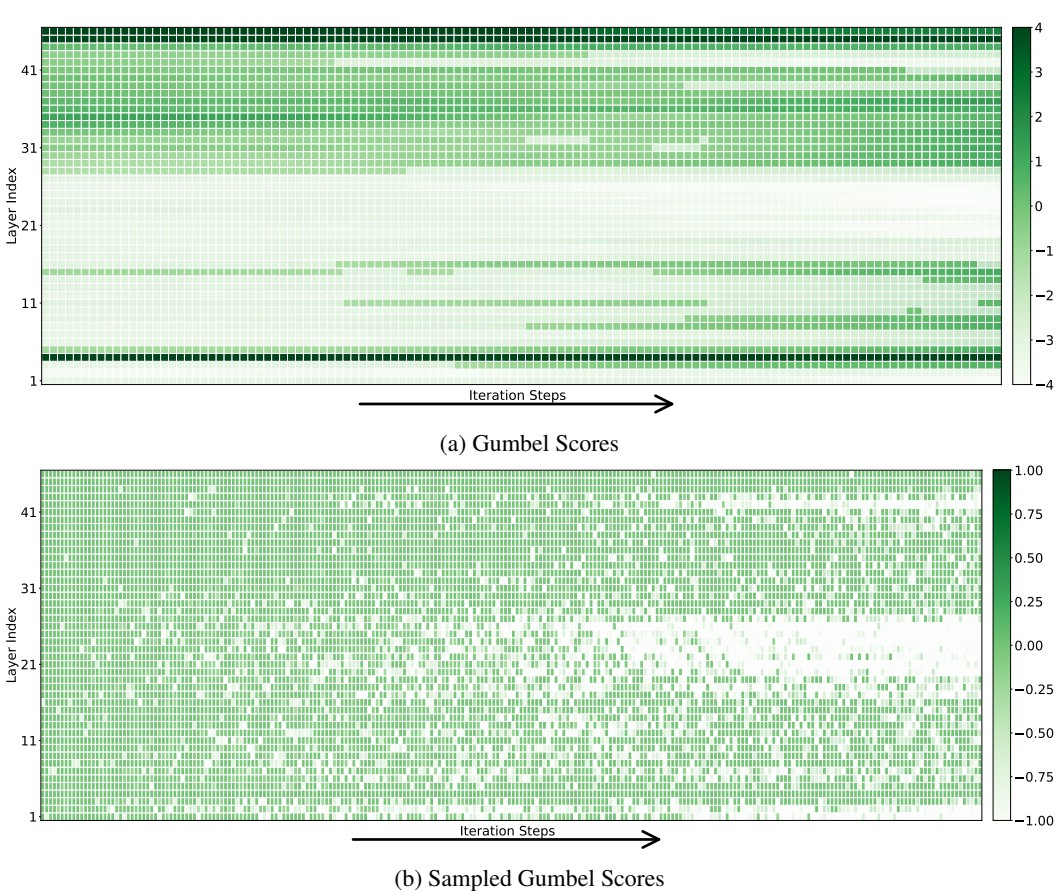

(a) Gumbel Scores

(b) Sampled Gumbel Scores

Figure 7: Gumbel scores optimization trajectory on Qwen2.5-14B-Instruct. (a): The Gumbel scores of each layer in each iteration step, the shadowed blocks indicate pruned layers. (b) The Sampled Gumbel scores during each forward pass.

Table 7: Hyperparameters details for pruning mask search and recovery training on LLaMA-2-7B, Qwen2.5-14B-Instruct, Qwen3-32B, DeepSeek-R1.

| Hyperparameters | LLaMA-2-7B | Qwen2.5-14B-Instruct | Qwen3-32B | DeepSeek-R1 |
|---|---|---|---|---|
| Pruning ratio | 60% | 50% | 12.5% / 25% | 12.5% / 25% |
| Learning rate | 1e-4 | 1e-4 | 1e-5 | 1e-4 |
| LR decay scheduler | Cosine | Cosine | Cosine | Cosine |
| LR warm-up steps | 60 | 60 | 60 | 120 |
| Training steps | 1200 | 1200 | 1200 | 2400 |
| Mask search steps | 120 | 120 | 120 | 240 |
| LoRA rank | - | - | - | 64 |
| LoRA alpha | - | - | - | 128 |
| Global batch size | 128 | 32 | 32 | 64 |
| Context length | 4096 | 16384 | 16384 | 4096 |
| Overall tokens | 0.5B | 0.5B | 0.5B | 0.5B |
| Distillation logits | Top 10 | Top 10 | Top 10 | Top 10 |
| Anneal coefficient $\beta$ | 0.9 | 0.9 | 0.9 | 0.9 |

