# OpenReview forum: "E$^3$-Pruner: Towards Efficient, Economical, and Effective Layer Pruning for Large Language Models"
_ICLR.cc/2026/Conference — Submitted to ICLR 2026_

### Official Review · Reviewer_6ac5 · 2025-10-25

**Soundness:** 2
**Presentation:** 2
**Contribution:** 2
**Rating:** 4
**Confidence:** 3

**Summary:**

The paper presents a layer pruning framework utilizing differentiable Gumbel-TopK sampler for and entropy-weighted knowledge distillation (KD) strategy for accuracy recovery. Experiments across different models show empirically better performance retention over baselines.

**Strengths:**

Efficient LLM pruning while addressing all three axes (accuracy, cost, speed) is of high importance. The experiments were well-designed in general and the results look promising.

**Weaknesses:**

+ The core technical contributions are straightforward heuristics adapted from existing differentiable pruning methods (Gumbel-Softmax) and weighting schemes. They offer negligible conceptual advance and are presented without necessary theoretical justification.

+ Lacking analysis of convergence and other key characteristics.

+ No wall-clock latency analysis.

+ The comparison restricts competitive baselines like DarwinLM.

The model optimization is presented as a black box, without any interpretability analysis of the resulting layer redundancy profile.

**Questions:**

+ Provide a rigorous hardware-level latency analysis on a standard GPU. Quantify the real-world inference throughput after pruning.

+ Explain why the Gumbel-TopK sampler should converge to a better mask than a simpler method, and show the sensitivity of the final mask composition to the annealing temperature (τ) schedule. Would a random search for layer indices yield a mask that performs comparably after distillation?

---

> ### Author Response · Authors · 2025-11-21
>
> > W1: The core technical contributions are straightforward heuristics adapted from existing differentiable pruning methods (Gumbel-Softmax) and weighting schemes. They offer negligible conceptual advance and are presented without necessary theoretical justification.
>
> **A1: First, our primary contribution lies in addressing a practical system challenge: enabling effective LLM pruning under real-world deployment constraints—namely, task effectiveness, training economy, and inference efficiency (Figure 2).** Despite extensive prior work, no existing method offers a unified, adaptive framework that simultaneously optimizes these three dimensions. Our approach bridges this gap by design.
>
> **Second, our differentiable Gumbel-TopK mechanism is distinct from Gumbel-Softmax.** While Gumbel-Softmax relaxes the argmax operation and—when used in pruning (e.g., ShearedLLaMA)—requires auxiliary regularization losses to induce sparsity, Gumbel-TopK directly relaxes the TopK operator, enabling end-to-end, auxiliary-loss-free layer selection. This is particularly well-suited for structured, multi-layer pruning in LLMs. As shown in our experiments, this design yields consistently superior performance.

---

> ### Author Response · Authors · 2025-11-21
>
> > W2: Lacking analysis of convergence and other key characteristics. The model optimization is presented as a black box, without any interpretability analysis of the resulting layer redundancy profile.
>
> **A2:Thank you for this insightful question. Due to space constraints, a detailed analysis of Gumbel-TopK optimization dynamics is provided in Appendix C (referenced in line 417).**
>
> Figure 7(b) shows that during training, the sampling distribution evolves from broad exploration across layers to sharp concentration on a stable subset—indicating convergence of the layer selection. Moreover, Figure 7(a) reveals that our differentiable updating refines initial importance estimates: some initially discarded layers are recovered, while previously overlooked redundant layers are pruned. This demonstrates that our method adaptively identifies redundancy beyond static heuristics.

---

> ### Author Response · Authors · 2025-11-21
>
> > W3: No wall-clock latency analysis.
>
> > Q1: Provide a rigorous hardware-level latency analysis on a standard GPU. Quantify the real-world inference throughput after pruning.
>
> **A3: We have already provided actual speedup ratios for wall-clock latency analysis in Tables 1 and 2.** Additionally, Appendix B.2 further provides detailed latency measurements for the pruned DeepSeek-R1 model deployed with vLLM, showing prefill and decode speedups across various batch sizes. The results confirm that layer pruning yields wall-clock acceleration that scales nearly proportionally with the compression ratio.

---

> ### Author Response · Authors · 2025-11-21
>
> > W4: The comparison restricts competitive baselines like DarwinLM.
>
> **A4: We directly use the official checkpoint released by DarwinLM and select the variant trained with a comparable (in fact, larger) token budget—1B tokens for DarwinLM versus 0.5B tokens for our method—to ensure a fair comparison.** Despite DarwinLM's multi-round evolutionary search and higher training cost, our method achieves consistently higher accuracy: 58.3% vs. 56.8% on LLaMA-2-7B and 61.9% vs. 59.4% on Qwen2.5-14B-Instruct. This demonstrates that our approach outperforms DarwinLM under more economical training conditions.

---

> ### Author Response · Authors · 2025-11-21
>
> > Q2: Explain why the Gumbel-TopK sampler should converge to a better mask than a simpler method.
>
> **A5: We provide a detailed explanation in Section 3.2, with the core insight lying in differentiable nature of Gumbel-TopK, which enables end-to-end, data-driven learning of the pruning mask.** While the most straightforward way to derive a mask from layer importance scores $S$ is via a hard TopK operation, this operation is non-differentiable and blocks gradient flow, preventing the scores from being updated via backpropagation.
>
> To address this, we employ the Gumbel-TopK relaxation, which renders the selection process differentiable. During forward passes, if the sampled mask erroneously removes a critical layer, the resulting loss increases; through the differentiable Gumbel-TopK pathway, gradients propagate back to the importance scores, suppressing scores of less important layers and reinforcing those of crucial ones. Over successive updates, this mechanism converges to an set of importance scores—and thus an better pruning mask.
>
> We further illustrate this dynamic evolution of the mask during training in Appendix C, where visualizations show how the mask gradually stabilizes toward a high-performing configuration.

---

> ### Author Response · Authors · 2025-11-21
>
> > Q3: Show the sensitivity of the final mask composition to the annealing temperature (τ) schedule.
>
> **A6: Thank you for the suggestion. We conducted an ablation study on the annealing temperature (τ) schedule on Qwen2.5-14B-Instruct, evaluating linear (used in the paper), exponential, and cosine decay schedules.** Results (shown below) indicate that our differentiable Gumbel-TopK search is robust to the choice of schedule—all variants consistently outperform existing pruning methods. We adopt the linear schedule for its simplicity and stable performance.
>
> ||SciQ|PIQA|WG|ArcE|ArcC\(25\)|HS\(10\)|LogiQA|BoolQ|MMLU\(5\)|AVG|
> |---|---|---|---|---|---|---|---|---|---|---|
> |E$^3$\-Pruner + Linear schedule|93.7|76.9|63.0|76.0|47.9|67.2|30.0|66.5|36.5|61.9|
> |E$^3$\-Pruner + Exponential schedule|93.7|75.8|62.5|74.6|48.4|65.5|27.8|66.7|36.9|61.3|
> |E$^3$\-Pruner + Cosine schedule|93.8|76.0|62.8|75.1|48.3|67.0|29.2|69.5|37.6|62.1|

---

> ### Author Response · Authors · 2025-11-21
>
> > Q4: Would a random search for layer indices yield a mask that performs comparably after distillation?
>
> **A7: Thank you for this insightful question. We conduct two additional experiments on Qwen2.5-14B-Instruct using random layer selection followed by our adaptive distillation strategy.** The results show that random pruning often removes critical layers, leading to significantly degraded performance—substantially worse than our method. This confirms that the layer mask learned via our differentiable Gumbel-TopK search is essential for high post-distillation accuracy.
>
> ||SciQ|PIQA|WG|ArcE|ArcC\(25\)|HS\(10\)|LogiQA|BoolQ|MMLU\(5\)|AVG|
> |---|---|---|---|---|---|---|---|---|---|---|
> |Random 1|89.2|71.4|58.3|69.6|39.8|56.2|26.9|60.1|26.7|55.4|
> |Random 2 |91.5|73.4|62.2|72.7|47.4|64.5|27.8|70.4|29.4|59.9|
> |E$^3$\-Pruner|93.7|76.9|63.0|76.0|47.9|67.2|30.0|66.5|36.5|61.9|

---

> > ### Comment · Reviewer_Kz3N · 2025-11-26
> >
> > Thank you for your reply. Based on your reply in A7, the final average performance of Random 2 and E$^3$-Pruner are 59.9 and 61.9 respectively, with a small difference in performance gap. However, random pruning does not require any search process, while your method requires a search, resulting in higher computational cost. Therefore, I believe this experiment does not fully demonstrate the effectiveness of the proposed method.

---

> > > ### Author Response · Authors · 2025-11-26
> > >
> > > > C1: Thank you for your reply. Based on your reply in A7, the final average performance of Random 2 and E$^3$-Pruner are 59.9 and 61.9 respectively, with a small difference in performance gap. However, random pruning does not require any search process, while your method requires a search, resulting in higher computational cost. Therefore, I believe this experiment does not fully demonstrate the effectiveness of the proposed method.
> > >
> > > A8: We appreciate the reviewer’s feedback and would like to clarify an important aspect of our experimental setup. **In all comparisons, we strictly maintain an identical total token budget of 0.5B tokens across both our method and the random baselines, ensuring a fair and controlled comparison.**
> > >
> > > Specifically, as stated in lines 424–425 of our manuscript, our E$^3$-Pruner allocates the first 10% of the total token budget (i.e., 0.05B tokens) to the searching stage and the remaining 90% (i.e., 0.45B tokens) to the fine-tuning stage. In contrast, the random baselines utilize the entire 0.5B tokens exclusively for fine-tuning, without any dedicated search or selection stage. Despite this more constrained allocation, our method consistently outperforms the random baselines.
> > >
> > > Furthermore, the results highlight a critical limitation of random selection: its performance is highly unstable. For instance, the “Random 1” variant achieves only 55.4% average accuracy—a substantial degradation compared to other runs and to our approach. In contrast, E$^3$-Pruner demonstrates robust and consistently superior performance across different experimental settings.
> > >
> > > We hope this clarification effectively addresses the your concerns regarding experimental fairness and methodological validity.

---

### Official Review · Reviewer_Kz3N · 2025-10-28

**Soundness:** 2
**Presentation:** 3
**Contribution:** 2
**Rating:** 4
**Confidence:** 4

**Summary:**

$E^{3}$-PRUNER is a layer pruning framework for large language models. This approach combines a differentiable search strategy using a Gumbel-TopK sampler with adaptive knowledge distillation techniques to find and train the optimal layer pruning configuration in an end-to-end manner. Extensive experiments on multiple models demonstrate the superiority of it over state-of-the-art approaches.

**Strengths:**

- The verification experiment is done quite thoroughly, spanning from 7B to 671B.
- The paper is easy to follow.
- The Gumbel-TopK sampler is introduced to replace the discrete TopK selection, achieving end-to-end differentiable optimization of layer pruning masks.

**Weaknesses:**

- As shown in Table 1, the model accuracy remains above 80% after pruning from 6.7B to 2.7B, which is extremely unusual in the field of LLM layer pruning. I am skeptical of this result and hope that the authors can provide a reproducible model checkpoint. In addition, since the proposed method can achieve such a high compression rate for LLaMA-2-7B, why is the compression rate so low for Qwen3-32B and DeepSeek-R1? In theory, the more model parameters there are, the more parts that can be compressed.
- Regarding the performance recovery of the pruned model, this paper uses adaptive knowledge distillation. Is it to fine-tune the entire model or only some parameters?
- Compared with traditional LoRA fine-tuning, how much performance improvement does the proposed adaptive knowledge distillation have, and how much difference is there in computational cost between these two?
- How much performance improvement would occur if other pruning methods were used with Adaptive Knowledge Distillation?
- No comparison with more advanced layer pruning methods.

**Questions:**

Given that you used knowledge distillation to restore the pruned DeepSeek-R1 model, how many GPUs do you use for the experiment?

---

> ### Author Response · Authors · 2025-11-21
>
> > W1: As shown in Table 1, the model accuracy remains above 80% after pruning from 6.7B to 2.7B, which is extremely unusual in the field of LLM layer pruning. I am skeptical of this result and hope that the authors can provide a reproducible model checkpoint. In addition, since the proposed method can achieve such a high compression rate for LLaMA-2-7B, why is the compression rate so low for Qwen3-32B and DeepSeek-R1? In theory, the more model parameters there are, the more parts that can be compressed.
>
> We appreciate the reviewer's questions and address them concisely below:
>
> **A1: High accuracy after aggressive pruning (6.7B to 2.7B)**
>
> This is primarily due to (i) post-pruning fine-tuning, which—consistent with prior work like Minitron [1]—can effectively recover performance with far fewer tokens than pretraining, and (ii) the use of relatively easy evaluation benchmarks (e.g., standard QA tasks). As noted in A3, performance degradation is more pronounced on harder reasoning tasks.
>
> **A2: The reprodicibility issue**
>
> In accordance with ICLR policy, we cannot share external links during the rebuttal period. Nevertheless, we will open-source the pruned models after the review period. In addtion, we have tried every effort to ensure reproducibility: for LLaMA-2-7B, we directly use publicly released checkpoints from ShearedLLaMA and DarwinLM, and explicitly specify the evaluation framework (lm-eval v0.4.8) in Section 4.1. These details allow any third party to reproduce our results exactly.
>
> **A3: Lower compression ratios for Qwen3-32B and DeepSeek-R1**
>
> These models are evaluated on complex reasoning tasks, which are highly sensitive to compression. Aggressive pruning often leads to catastrophic performance drops—a phenomenon also observed in recent works on large reasoning models (LRMs) under high sparsity [2]. Since our goal is practical deployment with minimal accuracy loss, we adopt more conservative compression ratios for these models.
>
> [1] Muralidharan, Saurav, et al. "Compact language models via pruning and knowledge distillation." Advances in Neural Information Processing Systems 37 (2024): 41076-41102.
>
> [2] Zhang, Nan, et al. "When reasoning meets compression: Benchmarking compressed large reasoning models on complex reasoning tasks." arXiv preprint arXiv:2504.02010 (2025).

---

> ### Author Response · Authors · 2025-11-21
>
> > W2: Regarding the performance recovery of the pruned model, this paper uses adaptive knowledge distillation. Is it to fine-tune the entire model or only some parameters?
>
> A4: We fine-tune the entire pruned model for most of our experiments. The only exception is DeepSeek-R1, where, due to NPU memory constraints, we employ LoRA for parameter-efficient fine-tuning.

---

> ### Author Response · Authors · 2025-11-21
>
> > W3: Compared with traditional LoRA fine-tuning, how much performance improvement does the proposed adaptive knowledge distillation have, and how much difference is there in computational cost between these two?
>
> **A5: Adaptive Knowledge Distillation is a loss function design, which is orthogonal and fully compatible with parameter-efficient fine-tuning methods such as LoRA.**
>
> To isolate its contribution, we compare adpative knowledge distillation against standard knowledge distillation in Table 4. Using the same pruned model and training setup, standard knowledge distillation achieves 60.8% accuracy, while adpative knowledge distillation further improves performance to 61.9%, demonstrating its effectiveness.

---

> ### Author Response · Authors · 2025-11-21
>
> > W4: How much performance improvement would occur if other pruning methods were used with Adaptive Knowledge Distillation?
>
> **A6: As noted in lines 300–301 of the paper, for all re-implemented baselines, we use identical training data and the same adaptive knowledge distillation protocol.** We also include the relevant results from Table 1 below for ease of reference. Under this strictly controlled and fair comparison, our E³-Pruner consistently outperforms other pruning methods, confirming that the performance gain originates from our pruning strategy itself.
>
> |||SciQ|PIQA|WG|ArcE|ArcC\(25\)|HS\(10\)|LogiQA|BoolQ|MMLU\(5\)|AVG|
> |---|---|---|---|---|---|---|---|---|---|---|---|
> |LLaMA\-2\-7B, 60% pruned |Minitron|86.4|67.2|56.5|58.0|32.9|49.4|29.3|62.4|\- |55.3|
> ||E$^3$\-Pruner|88.4  |71.2 |57.3 |65.4 |38.1 |54.6 |28.1 |63.6|\- |58.3|
> |Qwen2.5\-14B\-Instruct, 50% pruned|Minitron|91.9 |73.5 |62.0 |72.1 |44.9 |64.5 |28.4 |70.4 |32.8 |60.1|
> ||E$^3$\-Pruner|93.7    |76.9|63.0|76.0|47.9 |67.2|30.0 |66.5 | 36.5 |61.9|

---

> ### Author Response · Authors · 2025-11-21
>
> > W5: No comparison with more advanced layer pruning methods.
>
> **A7: We have comprehensively compared against recent fine-tuning-based layer pruning methods—including ShearedLLaMA (2024), Minitron (2024), and DarwinLM (2025)—using identical training budgets to ensure a fair evaluation.** Training-free approaches (e.g., SLEB, Shortened LLaMA) exhibit severe accuracy degradation and are thus impractical for real-world deployment; we include only ShortGPT as a representative example of this limitation.
>
> To further address this concern, we evaluate additional recent pruning methods on Qwen2.5-14B-Instruct, both with and without our adaptive distillation strategy. As shown below, E³-Pruner consistently outperforms all baselines, highlighting the efficacy of our approach.
>
> ||SciQ|PIQA|WG|ArcE|ArcC\(25\)|HS\(10\)|LogiQA|BoolQ|MMLU\(5\)|AVG|
> |---|---|---|---|---|---|---|---|---|---|---|
> |Shortened LLaMA|59.8|53.9|51.4|34.4|22.6|26.1|25.5|45.5|25.6|38.3|
> |Shortened LLaMA + Adaptive KD|91.0|75.6|59.6|74.7|47.5|64.8|27.3|60.8|31.6|59.2|
> |SLEB|75.0|64.2|51.5|50.9|24.9|42.1|25.5|58.4|24.6|46.3|
> |SLEB + Adaptive KD|93.0|76.8|61.3|75.4|47.8|66.6|28.6|65.1|33.7|60.9|
> |**E$^3$\-Pruner**|**93.7**|**76.9**|**63.0**|**76.0**|**47.9**|**67.2**|**30.0**|**66.5**|**35.8**|**61.9**|

---

> ### Author Response · Authors · 2025-11-21
>
> > Q: Given that you used knowledge distillation to restore the pruned DeepSeek-R1 model, how many GPUs do you use for the experiment?
>
> **A8: For the 671B DeepSeek-R1 model, we perform efficient LoRA fine-tuning using 64 NPUs (64 GB memory each)—a configuration detailed in Appendix D due to space constraints.**
>
> Specifically, we extend the PEFT library to support LoRA adaptation of DeepSeek-R1's MoE architecture and integrate it with Megatron's tensor parallelism (TP) and expert parallelism (EP). This integration enables full-parameter LoRA fine-tuning across all model components while staying within memory limits. With a LoRA rank of 128 and TP=8/EP=8, the trainable parameters amount to less than 3% of the full model.
>
> As a result, the entire pruning and fine-tuning pipeline for DeepSeek-R1 completes in under 24 hours, representing a significant engineering advance for ultra-large MoE models. We plan to open-source our implementation to promote reproducibility and broader community adoption.

---

### Official Review · Reviewer_iaGe · 2025-11-02

**Soundness:** 3
**Presentation:** 3
**Contribution:** 3
**Rating:** 6
**Confidence:** 4

**Summary:**

This paper proposes E3-PRUNER, which is a layer-pruning framework for large language models that learns which layers to keep or remove through a differentiable mask optimized with a Gumbel-TopK sampler. It further uses entropy-aware adaptive knowledge distillation to retain performance. Experiments show it achieves up to 2.18× speedup with minimal accuracy loss, offering an efficient, economical, and effective solution for LLM compression.

**Strengths:**

1. This paper proposes a differentiable mask learning framework (Gumbel-TopK) for layer pruning, enabling efficient gradient-based layer selection.
2. The method achieves a good pruned model performance, outperforming prior pruning methods.
3. The paper further introduces entropy-aware adaptive knowledge distillation, effectively preserving key reasoning tokens.
4. The experiments demonstrate consistent and superior results across multiple LLMs with minimal accuracy loss.

**Weaknesses:**

1. The paper provides a limited theoretical explanation of why the Gumbel-TopK mask search is able to identify the optimal layers.
2. The paper does not clarify whether the performance gain comes from the layer pruning method or the Adaptive Knowledge Distillation. It would be better to compare the zero-shot performance of the pruned model without fine-tuning or apply Adaptive KD to baseline pruning methods to evaluate their relative effects.

**Questions:**

This is the first academic paper I have seen that fine-tunes the 671B DeepSeek-R1 model. How many GPUs do you use? This information could be included in the Settings section of the paper.

---

> ### Author Response · Authors · 2025-11-21
>
> > W1: The paper provides a limited theoretical explanation of why the Gumbel-TopK mask search is able to identify the optimal layers.
>
> **A1: We provide a detailed explanation in Section 3.2, with the core insight lying in differentiable nature of Gumbel-TopK, which enables end-to-end, data-driven learning of the pruning mask.** While the most straightforward way to derive a mask from layer importance scores $S$ is via a hard TopK operation, this operation is non-differentiable and blocks gradient flow, preventing the scores from being updated via backpropagation.
>
> To address this, we employ the Gumbel-TopK relaxation, which renders the selection process differentiable. During forward passes, if the sampled mask erroneously removes a critical layer, the resulting loss increases; through the differentiable Gumbel-TopK pathway, gradients propagate back to the importance scores, suppressing scores of less important layers and reinforcing those of crucial ones. Over successive updates, this mechanism converges to an set of importance scores—and thus an better pruning mask.
>
> We further illustrate this dynamic evolution of the mask during training in Appendix C, where visualizations show how the mask gradually stabilizes toward a high-performing configuration.

---

> ### Author Response · Authors · 2025-11-21
>
> > W2: The paper does not clarify whether the performance gain comes from the layer pruning method or the Adaptive Knowledge Distillation. It would be better to compare the zero-shot performance of the pruned model without fine-tuning or apply Adaptive KD to baseline pruning methods to evaluate their relative effects.
>
> **A2: We appreciate the reviewer's insightful comment. While our approach jointly include both pruning and fine-tuning, below we show that their individual roles are also helpful.** We evaluate the Qwen2.5-14B-Instruct model pruned 50% layers at two key stages: (i) immediately after stage 1 (mask searching) and (ii) after the full recovery stage 2 (fine-tuning). For comparison, we also include the zero-shot performance of ShortGPT (no fine-tuning) as follows.  After stage 1—using only 0.05B training tokens—the pruned model achieves an average accuracy of 55.5%, a +18.4% gain over ShortGPT (37.1%). And the adaptive knowledge distillation in stage 2 further improves accuracy by +6.4% points, reaching 61.9%.  These results indicate that both our differentiable pruning strategy (stage 1) and adaptive distillation (stage 2) play critical and complementary roles in recovering and enhancing performance.
>
> ||SciQ|PIQA|WG|ArcE|ArcC\(25\)|HS\(10\)|LogiQA|BoolQ|MMLU\(5\)|AVG|Token budget|
> |---|---|---|---|---|---|---|---|---|---|---|---|
> |ShortGPT|31.6|56.7|50.3|30.5|27.6|34.0|24.9|53.8|24.6|37.1|0.01B|
> |E$^3$\-Pruner stage1|88.3|72.0|54.5|69.2|40.9|57.0|26.9|62.0|28.4|55.5|0.05B|
> |E$^3$\-Pruner stage2|93.7|76.9|63.0|76.0|47.9|67.2|30.0|66.5|35.8|61.9|0.5B|

---

> ### Author Response · Authors · 2025-11-21
>
> > Q: This is the first academic paper I have seen that fine-tunes the 671B DeepSeek-R1 model. How many GPUs do you use? This information could be included in the Settings section of the paper.
>
> **A3: Thank you for the recognition. For the 671B DeepSeek-R1 model, we perform efficient LoRA fine-tuning using 64 NPUs (64 GB memory each)—a configuration detailed in Appendix D due to space constraints.**
>
> Specifically, we extend the PEFT library to support LoRA adaptation of DeepSeek-R1's MoE architecture and integrate it with Megatron's tensor parallelism (TP) and expert parallelism (EP). This integration enables full-parameter LoRA fine-tuning across all model components while staying within memory limits. With a LoRA rank of 128 and TP=8/EP=8, the trainable parameters amount to less than 3% of the full model.
>
> As a result, the entire pruning and fine-tuning pipeline for DeepSeek-R1 completes in under 24 hours, representing a significant engineering advance for ultra-large MoE models. We plan to open-source our implementation to promote reproducibility and broader community adoption.

---

### Official Review · Reviewer_Gw2L · 2025-11-02

**Soundness:** 2
**Presentation:** 3
**Contribution:** 2
**Rating:** 2
**Confidence:** 5

**Summary:**

This paper proposes $E^3$-Pruner, a layer pruning framework for large language models that integrates a differentiable Gumbel-TopK sampler for efficient mask optimization and an entropy-aware adaptive knowledge distillation strategy to address the limitations of existing methods in performance preservation, training cost, and inference acceleration.

**Strengths:**

1. This paper combines a differentiable Gumbel-TopK sampler, for efficient and accurate pruning mask search with an entropy-aware adaptive knowledge distillation strategy, for enhanced knowledge transfer with reduced computational cost.
2. This paper executes extensive experiments on across diverse LLMs with different sizes and architectures, and evaluates on multiple benchmarks, demonstrating the generalization and practicality of the proposed $E^3$-Pruner framework.

**Weaknesses:**

1. **Limited novelty:** Gumbel-TopK sampling for pruning has been explored in prior works [1-2] for model compression, and the progressive layer pruning strategy is a common approach (e.g., SLEB [3]) with no significant innovation here.
2. **Incomplete baselines:** Heuristic layer pruning methods like SLEB [3] and Shortened LLaMA [4] should be added as baselines to enable more comprehensive comparison and better highlight the proposed method’s advantages.
3. **Unfair comparison design:** The paper fails to clarify whether baseline methods update parameters (for performance recovery after pruning). Comparing parameter-updating and non-updating methods (e.g., training-free ShortGPT’s pruned model vs. $E^3$-Pruner’s fine-tuned model) is inappropriate; training-free baselines should be compared with $E^3$-Pruner’s post-search model.
4. **Insufficient evidence for training economy:** Only training token counts are reported. Actual training time and computational resources (e.g., GPU hours) during pruning should be provided to substantiate the claim of training efficiency.
5. **Lack of ablation study on layer importance initialization:** Comparing different initialization methods (e.g., random initialization, ShortGPT’s layer importance metric) would clarify how initialization impacts final performance.

[1] Gonzalez-Carabarin, et al. Dynamic Probabilistic Pruning: A General Framework for Hardware-Constrained Pruning at Different Granularities. TNNLS 2022.

[2] Tan, et al. Mutually-aware Sub-Graphs Differentiable Architecture Search. arXiv 2021.

[3] Song, et al. SLEB: Streamlining LLMs through Redundancy Verification and Elimination of Transformer Blocks. ICML 2024.

[4] Kim, et al. Shortened LLaMA: Depth Pruning for Large Language Models with Comparison of Retraining Methods. arXiv 2024.

**Questions:**

1. Minitron proposed iterative prune+distill for model pruning. How many prune+distill iterations were conducted in your experiments? If multiple, please provide the number of iterations and detailed settings; if only one, explain the rationale for choosing single over multiple iterations.
2. How does the performance of ShortGPT-pruned models fine-tuned with $E^3$-Pruner’s adaptive knowledge distillation compare to pruned models after $E^3$-Pruner's two stages process? This comparison would help isolate the effects of the pruning method and distillation strategy.
3. Could you provide qualitative examples of outputs from the original and pruned models on specific tasks (e.g. MMLU using Qwen2.5-14B-Instruct)? This would illustrate the practical impact of pruning on model performance.

---

> ### Author Response · Authors · 2025-11-21
>
> > W1: Limited novelty: Gumbel-TopK sampling for pruning has been explored in prior works [1-2] for model compression, and the progressive layer pruning strategy is a common approach (e.g., SLEB [3]) with no significant innovation here.
>
> A1: We appreciate the reviewer's feedback. Below we clarify the novelty of our work in two points:
> * **The significance of this work**
>
>   Pruning large language models (LLMs) under practical deployment constraints—task effectiveness, training economy, and inference efficiency (as highlighted in Figure 2)—remains a critical and unsolved challenge. While neural network compression has evolved over more than a decade, most classical techniques do not directly transfer to LLMs due to their scale, autoregressive structure, and inference cost profile. Even recently, representative LLM compression works largely repurpose established strategies: for instance, ShearedLLaMA adopts trainable soft masks and regularization from prior work [3], while DarwinLM relies on evolutionary search—a well-established paradigm in neural architecture search (NAS). However, none of these methods address the core trade-offs in a unified and adaptive framework and resolve the real-world efficiency bottlenecks for modern LLMs, which is precisely what our approach aims to achieve.
>
>
> * **Technical distinctions and innovations**
>
>   Our work fundamentally differs from prior uses of Gumbel-TopK:
>   * [1] applies it to unstructured weight pruning in CNNs—orthogonal to our structured layer\-wise pruning for LLMs.
>   * [2] uses it for neural architecture search over heterogeneous operators \(e.g., skip connections, convolutions\), which concerns architecture design, not compression.
>
>   While progressive pruning is a standard heuristic, our core innovation is a two\-stage LLM layer pruning framework:
>   * Differentiable mask optimization : A Gumbel-TopK–based mechanism that enables end-to-end training of pruning masks, facilitating efficient, precise, and adaptive selection of pruning mask—critical for preserving capabilities of LLMs
>   * Adaptive knowledge distillation: We propose a tailored distillation strategy that dynamically aligns the student model's learning with the most informative signals from the teacher. This allows for rapid performance recovery with minimal additional training cost, which is essential for practical deployment.
>
>   To the best of our knowledge, this methodology has not been explored in prior literature.
>
>
>
> [1] Gonzalez-Carabarin, et al. Dynamic Probabilistic Pruning: A General Framework for Hardware-Constrained Pruning at Different Granularities. TNNLS 2022.
>
> [2] Tan, et al. Mutually-aware Sub-Graphs Differentiable Architecture Search. arXiv 2021.
>
> [3] Louizos, Christos, Max Welling, and Diederik P. Kingma. "Learning sparse neural networks through $ L_0 $ regularization." arXiv preprint arXiv:1712.01312 (2017).

---

> ### Author Response · Authors · 2025-11-21
>
> > W2: Incomplete baselines: Heuristic layer pruning methods like SLEB [3] and Shortened LLaMA [4] should be added as baselines to enable more comprehensive comparison and better highlight the proposed method’s advantages.
>
> **A2: Thanks for the suggestion. In the table below, we show that heruistic layer pruning methods (SLEB, Shortened LLaMA) both without and with fine-tuning cannot surpass the proposed E$^3$-Pruner.** The results are based on Qwen2.5-14B-Instruct with 50% layers pruned. This gap arises because these methods rely on static importance heuristics that fail to adapt to task-specific redundancy, whereas our differentiable search yields a more accurate pruning mask. These results underscore the effectiveness and adaptivity of our approach.
>
> ||SciQ|PIQA|WG|ArcE|ArcC\(25\)|HS\(10\)|LogiQA|BoolQ|MMLU\(5\)|AVG|
> |---|---|---|---|---|---|---|---|---|---|---|
> |Shortened LLaMA|59.8|53.9|51.4|34.4|22.6|26.1|25.5|45.5|25.6|38.3|
> |Shortened LLaMA + Adaptive KD|91.0|75.6|59.6|74.7|47.5|64.8|27.3|60.8|31.6|59.2|
> |SLEB|75.0|64.2|51.5|50.9|24.9|42.1|25.5|58.4|24.6|46.3|
> |SLEB + Adaptive KD|93.0|76.8|61.3|75.4|47.8|66.6|28.6|65.1|33.7|60.9|
> |**E$^3$\-Pruner**|**93.7**|**76.9**|**63.0**|**76.0**|**47.9**|**67.2**|**30.0**|**66.5**|**35.8**|**61.9**|

---

> ### Author Response · Authors · 2025-11-21
>
> > W3: Unfair comparison design: The paper fails to clarify whether baseline methods update parameters (for performance recovery after pruning). Comparing parameter-updating and non-updating methods (e.g., training-free ShortGPT’s pruned model vs. $E^3$-Pruner’s fine-tuned model) is inappropriate; training-free baselines should be compared with $E^3$-Pruner’s post-search model.
>
> **A3: Thank you for raising this important point.  We include ShortGPT only to illustrate that training-free pruning is impractical due to severe accuracy degradation (Figure 2). All other baselines—whether official models or our re-implementations—use post-pruning fine-tuning comparable to our method.**
> Our evaluation ensures a thorough and fair comparison: we report accuracy, actual speedup, and training token budget for every method under identical settings (Section 4.1). Among training-based approaches, E$^3$-Pruner consistently achieves higher accuracy and greater speedup at the comparable or even less token cost, demonstrating superior effectiveness and efficiency.

---

> ### Author Response · Authors · 2025-11-21
>
> > W4: Insufficient evidence for training economy: Only training token counts are reported. Actual training time and computational resources (e.g., GPU hours) during pruning should be provided to substantiate the claim of training efficiency.
>
> **A4: While training token count offers a fair, platform-agnostic measure of training cost, actual runtime and resource usage vary significantly with implementation, hardware, and framework—rendering direct comparisons unreliable.** This is especially true for official baseline checkpoints, where differences in training infrastructure and codebases make runtime-based cost assessments uninformative. This reflects an objective practical challenge, not a limitation of our study.
>
> Nevertheless, for transparency, we report in Appendix D that our full pipeline for Qwen2.5-14B-Instruct (pruning + fine-tuning) completes in ~1.5 hours on 64 NPUs.

---

> ### Author Response · Authors · 2025-11-21
>
> > W5: Lack of ablation study on layer importance initialization: Comparing different initialization methods (e.g., random initialization, ShortGPT’s layer importance metric) would clarify how initialization impacts final performance.
>
> **A5: We have already conducted an ablation study on layer importance initialization in Section 4.3. As shown in Figure 5, our initialization strategy consistently improves accuracy across datasets.** Specifically, the first and second bars in Figure 5 compare models trained with the same data but initialized using ShortGPT's layer importance metric versus our proposed strategy, respectively. The results demonstrate that our initialization yields consistent and meaningful gains in performance.

---

> ### Author Response · Authors · 2025-11-21
>
> > Q1: Minitron proposed iterative prune+distill for model pruning. How many prune+distill iterations were conducted in your experiments? If multiple, please provide the number of iterations and detailed settings; if only one, explain the rationale for choosing single over multiple iterations.
>
> **A6: We perform only a single iteration throughout the entire pruning process. In contrast, multi-iteration strategies (e.g., Minitron) require 2–4 rounds of repeated training and evaluation, incurring significantly higher computational costs that conflict with our goal of training economy.**
>
> Instead, we employ a progressive pruning schedule, gradually reducing the layer retention budget $k$ during search. This enables effective layer selection in one pass, cutting total training cost by 2–4× versus multi-iteration approaches while maintaining competitive performance.

---

> ### Author Response · Authors · 2025-11-21
>
> > Q2: How does the performance of ShortGPT-pruned models fine-tuned with $E^3$-Pruner’s adaptive knowledge distillation compare to pruned models after $E^3$-Pruner's two stages process? This comparison would help isolate the effects of the pruning method and distillation strategy.
>
> A7: Thanks for the suggestion. Acutally, In Table 1, we compare against the **Minitron strategy—which uses the same BI-based layer importance as ShortGPT and an identical distillation token budget**—and show clear gains: 58.3% vs. 55.3% on LLaMA-2-7B (60% pruning) and 61.9% vs. 60.1% on Qwen2.5-14B-Instruct (50% pruning), as also extracted to the table below. We will clarify this fair comparison more explicitly in the revised manuscript.
>
> |||SciQ|PIQA|WG|ArcE|ArcC\(25\)|HS\(10\)|LogiQA|BoolQ|MMLU\(5\)|AVG|
> |---|---|---|---|---|---|---|---|---|---|---|---|
> |LLaMA\-2\-7B, 60% pruned |Minitron|86.4|67.2|56.5|58.0|32.9|49.4|29.3|62.4|\- |55.3|
> ||E$^3$\-Pruner|88.4  |71.2 |57.3 |65.4 |38.1 |54.6 |28.1 |63.6|\- |58.3|
> |Qwen2.5\-14B\-Instruct, 50% pruned|Minitron|91.9 |73.5 |62.0 |72.1 |44.9 |64.5 |28.4 |70.4 |32.8 |60.1|
> ||E$^3$\-Pruner|93.7    |76.9|63.0|76.0|47.9 |67.2|30.0 |66.5 | 36.5 |61.9|

---

> ### Author Response · Authors · 2025-11-21
>
> > Q3: Could you provide qualitative examples of outputs from the original and pruned models on specific tasks (e.g. MMLU using Qwen2.5-14B-Instruct)? This would illustrate the practical impact of pruning on model performance.
>
> **A8: Thank you for this insightful suggestion. We provide qualitative comparisons of the original and pruned Qwen2.5-14B-Instruct on MMLU using a 0-shot prompt (examples below).**
>
> |Prompt|Original model response|Pruned model response|
> |---|---|---|
> |The following are multiple choice questions \(with answers\) about high school mathematics.\\n\\nA positive integer n is called “powerful” if, for every prime factor p of n, p^2 is also a factor of n. An example of a powerful number is\\nA. 392\\nB. 336\\nC. 300\\nD. 297\\nAnswer:| A| A|
> |The following are multiple choice questions \(with answers\) about abstract algebra.\\n\\nFind the degree for the given field extension Q\(sqrt\(2\), sqrt\(3\), sqrt\(18\)\) over Q.\\nA. 0\\nB. 4\\nC. 2\\nD. 6\\nAnswer:| B| B. 4|
> |The following are multiple choice questions \(with answers\) about college computer science.\\n\\nConsider the collection of all undirected graphs with 10 nodes and 6 edges. Let M and m, respectively, be the maximum and minimum number of connected components in any graph in the collection. If a graph has no selfloops and there is at most one edge between any pair of nodes, which of the following is true?\\nA. M = 10, m = 10\\nB. M = 10, m = 1\\nC. M = 7, m = 4\\nD. M = 6, m = 4\\nAnswer:| C| A|
>
> **Given the brevity of multiple-choice answers in MMLU, we additionally include examples which requiring longer, free-form responses. As shown, while the pruned model produces different outputs from the original, its responses remain fluent and semantically correct—demonstrating its practical viability for real-world deployment.**
>
> |Prompt|Original model response|Pruned model response|
> |---|---|---|
> |The president of the United States is|not only a leader, but also an icon. As such, his appearance is often scrutinized and analyzed by the public. One aspect of a president's appearance that has been discussed in recent years is their height.\\nAccording to data from the White House Historical Association, the average height of U.S. presidents is 5 feet 10 inches \(178 cm\). However, there have been some notable exceptions. For example, James Madison was the shortest president at 5 feet 4 inches.|the head of state and head of government of the United States. The president is the commander\-in\-chief of the United States Armed Forces and is the chief diplomat of the United States. The president is elected by the Electoral College to a four\-year term and can be re\-elected only once. The president is the only person who can be impeached by the House of Representatives and removed from office by the Senate. The president is also the only person who can grant pardons and reprieves.|
> |The capital of France is|Paris. It is also the largest city in the country and one of the most popular tourist destinations in Europe. The population of Paris is around 2.2 million people, but the metropolitan area is home to about 12 million people.\\nParis is located in northern France along the Seine River. It is a major economic center for the European Union and the world. It is also a hub for culture and arts, with many museums, galleries, and theaters.|Paris. The country is located in Western Europe and is surrounded by the Atlantic Ocean, the English Channel, and the Mediterranean Sea. France is the most visited country in the world, with 82 million tourists a year. The country is also the most popular tourist destination in the world, with 80 million tourists a year. The country is also the most popular tourist destination in the world, with 80 million tourists a year.|
> |The future of AI is|here, and it’s called “Artificial General Intelligence” or AGI. Unlike current AI systems that can only perform specific tasks, AGI is designed to think, learn, and reason like a human being. While the idea of AGI may sound like something out of science fiction, experts believe it could be just around the corner.\\nOne of the key benefits of AGI is its ability to adapt and learn from new information.|here, and it’s already changing the way we live and work. From self\-driving cars to virtual assistants, AI is becoming increasingly integrated into our daily lives. But what does the future of AI look like, and how will it impact society? In this article, we’ll explore the potential of AI and its impact on society, as well as the ethical considerations and challenges that come with it.\\nThe future of AI is already here, and it’s changing the way we live and work.|

---

> > ### Comment · Reviewer_Gw2L · 2025-11-27
> >
> > Thank you for your reply. While I appreciate your response, I still have a few concerns.
> > 1. As you mentioned in A7, Minitron strategy uses the same BI-based layer importance as ShortGPT, which means ShortGPT +   Adaptive KD. However, Minitron pruning more structures than transformer layers, including attention head and channels number, etc. It is recommended to clarify the configuration of Minitron to avoid unnecessary misunderstandings.
> >
> > 2. A key feature of Minitron is the iterative pruning and distillation approach. Only a single iteration does not align with Minitron strategy. It remains possible to keep the overall budget unchanged while splitting it evenly across all rounds. Moreover, It is also unclear which strategy is superior: the "pruning-distillation-pruning iterative" strategy (i.e., the Minitron strategy) or the "pruning-then-distillation once" strategy (i.e., proposed in this paper).
> >
> > 3. The performance improvement is marginal, regarding $E^3$-Pruner v.s. SLEB + Adaptive KD in A2. It may not provide strong evidence that **the propose Gumbel-TopK sampler for progressive layer pruning** is superior to **SLEB's heuristic importance metric based progressive layer pruning**. Also, it is suggested that the performance of $E^3$-Pruner without Adaptive KD is included for comprehensive comparision.

---

> ### Author Response · Authors · 2025-11-29
>
> > C1: As you mentioned in A7, Minitron strategy uses the same BI-based layer importance as ShortGPT, which means ShortGPT + Adaptive KD. However, Minitron pruning more structures than transformer layers, including attention head and channels number, etc. It is recommended to clarify the configuration of Minitron to avoid unnecessary misunderstandings.
>
> **A9: Thank you for your insightful suggestion. In our experiments, we re\-implement the Minitron strategy under a layer\-pruning\-only setting to ensure a fair comparison with other baselines.** Specifically, we apply only layer pruning and do not prune other structures such as attention heads or MLP channels. This configuration is explicitly detailed in  Baselines of Section 4.1 in the paper. We sincerely appreciate your suggestion and will further clarify this point in the revised manuscript to enhance the transparency of our experimental setup.
>
> > C2: A key feature of Minitron is the iterative pruning and distillation approach. Only a single iteration does not align with Minitron strategy. It remains possible to keep the overall budget unchanged while splitting it evenly across all rounds. Moreover, It is also unclear which strategy is superior: the "pruning-distillation-pruning iterative" strategy (i.e., the Minitron strategy) or the "pruning-then-distillation once" strategy (i.e., proposed in this paper).
>
> **A10: We would like to clarify that the choice between iterative versus single-round pruning-distillation is largely orthogonal to our core contribution and can be flexibly integrated.** In this work, our primary focus is on optimizing task effectiveness, training economy, and inference efficiency under practical deployment constraints. To this end, we adopt a single\-stage pruning followed by distillation strategy for its training efficiency and lower resource footprint.
>
> That said, our framework is readily extensible to iterative schemes \(e.g., Minitron's pruning–distillation–pruning cycles\) when budget allows. While a comprehensive comparison between iterative and non\-iterative strategies is beyond our paper’s scope—since we aim not to advocate a specific distillation schedule but to deliver an efficient, deployment\-oriented solution—we provide empirical results combining our method with Minitron's 2\- and 4\-iteration protocols. As shown, our E³\-Pruner consistently outperforms Minitron across all settings. Notably, even our one\-shot variant surpasses all iterative Minitron variants, underscoring the efficacy of our Gumbel\-TopK sampler in identifying high\-quality pruning masks.
>
> ||SciQ|PIQA|WG|ArcE|ArcC\(25\)|HS\(10\)|LogiQA|BoolQ|MMLU\(5\)|AVG|
> |---|---|---|---|---|---|---|---|---|---|---|
> |Minitron \(1 iteration\)|91.9|73.5|62.0|72.1|44.9|64.5|28.4|70.4|32.8|60.1|
> |Minitron \(2 iterations\)|93.2|76.2|63.7|73.9|45.7|65.4|30.1|65.2|35.1|60.9|
> |Minitron \(4 iterations\)|93.6|74.8|62.6|75.3|46.5|66.1|29.0|67.2|37.7|61.4|
> ||||||||||||
> |E$^3$-Pruner \(1 iteration\)|93.7|76.9|63.0|76.0|47.9|67.2|30.0|66.5|35.8|61.9|
> |E$^3$-Pruner \(2 iterations\)|94.1|75.5|62.9|75.0|47.5|66.0|30.0|68.1|40.0|62.1|
> |E$^3$-Pruner \(4 iterations\)|93.4|75.5|62.7|74.9|51.1|66.7|30.3|69.1|40.1|62.6|

---

> ### Author Response · Authors · 2025-11-29
>
> > C3: The performance improvement is marginal, regarding $E^3$-Pruner v.s. SLEB + Adaptive KD in A2. It may not provide strong evidence that the propose Gumbel-TopK sampler for progressive layer pruning is superior to SLEB's heuristic importance metric based progressive layer pruning. Also, it is suggested that the performance of $E^3$-Pruner without Adaptive KD is included for comprehensive comparision.
>
> A11: **We would like to emphasize that, while fine-tuning reduces the performance gap among approaches, E³\-Pruner consistently outperforms SLEB both before and after Adaptive KD fine\-tuning, despite operating under a smaller total token budget**. To illustrate this, we provide a detailed comparison of performance before and after Adaptive_KD for ShortGPT, SLEB and E3\-Pruner as below.  As shown,  E³-Pruner maintains the highest accuracy throughout, underscoring the effectiveness of our Gumbel-TopK-based sampler in identifying a superior pruning mask compared to SLEB's heuristic importance metric.
>
> Moreover, although both SLEB and E³\-Pruner consume 0.5B tokens for training \(0.05B for searching and 0.45B for fine\-tuning in E³\-Pruner\), SLEB incurs an additional 0.5B tokens for heuristic importance calibration—a process involving O\(N²\) candidate evaluations, as documented in the original SLEB paper. Consequently, E³\-Pruner achieves superior performance under a strictly smaller total token budget, further demonstrating its efficiency and effectiveness.
>
> ||SciQ|PIQA|WG|ArcE|ArcC\(25\)|HS\(10\)|LogiQA|BoolQ|MMLU\(5\)|AVG|
> |---|---|---|---|---|---|---|---|---|---|---|
> |ShortGPT|31.6|56.7|50.3|30.5|27.6|34.0|24.9|53.8|24.6|37.1|
> |SLEB|75.0|64.2|51.5|50.9|24.9|42.1|25.5|58.4|24.6|46.3|
> |E$^3$-Pruner stage1|88.3|72.0|54.5|69.2|40.9|57.0|26.9|62.0|28.4|55.5|
> ||||||||||||
> |ShortGPT + Adaptive KD|91.9|73.5|62.0|72.1|44.9|64.5|28.4|70.4|32.8|60.1|
> |SLEB + Adaptive KD|93.0|76.8|61.3|75.4|47.8|66.6|28.6|65.1|33.7|60.9|
> |E$^3$-Pruner stage2|93.7|76.9|63.0|76.0|47.9|67.2|30.0|66.5|35.8|61.9|

---

### Official Review · Reviewer_gvMZ · 2025-11-03

**Soundness:** 3
**Presentation:** 3
**Contribution:** 2
**Rating:** 6
**Confidence:** 5

**Summary:**

This paper proposed E³-PRUNER, a two-stage layer-wise pruning framework tailored for LLMs. In the mask search stage, it learns a binary layer mask with a Gumbel-TopK sampler and STE backward, warmed up by KL-based layer-importance initialization and a curriculum schedule that gradually increases the pruning ratio. In the recovery stage, it applies adaptive knowledge distillation, using offline teacher Top-K logits and token-wise entropy weighting on the per-token loss. In the experiments, the method preserves accuracy while reducing depth and improving latency under modest training budgets across models and tasks.

**Strengths:**

1. The problem statement and challenge are clear and significant. Section 2.2 provides a detailed analysis of why existing training-free, differentiable, and NAS-based approaches each fall short (e.g., accuracy drop, high token budgets, irregular speedups), setting up a precise target for improvement.
2. This paper is fairly written. The narrative is well-structured and easy to follow: motivation, formulation, differentiable mask search, and recovery via adaptive KD. Figures and algorithms (Fig. 3; Algs. 1–2; Eqs. (4)–(6)) make the procedure executable and reduce ambiguity about forward/backward behavior and training schedules.
3. Clear real-world deployment evidence supports the author’s claim on effectiveness and inference efficiency gain. Beyond accuracy tables, the paper reports wall-clock improvements for large-scale models

**Weaknesses:**

1. The behavioral consistency metric is weakly defined. The paper measures consistency as average accuracy on a small mixed set rather than teacher–student agreement (e.g., output match rate, output distributions, or log-prob correlations). This undermines the claim that KD better preserves behavior.

2. Storage/IO for offline KD is unquantified. The method relies on offline Top-K logits and asserts “minor storage,” using Top-10 in all configs, but provides no concrete footprint or bandwidth numbers under long contexts and 0.5B-token training budgets. This limits reproducibility and deployability assessments.

3. Fairness of the comparison. ShortGPT is training-free, so its lower accuracy vs. trained methods is expected; using it as a headline accuracy comparator can be misleading. A fair alternative is ShortGPT with the same recovery budget (SFT/KD) to normalize token use.

4. Missing comparisons to recent depth-pruning baselines. The paper does not compare to other layer/depth pruners that replace blocks via KD (e.g., LLM-Streamline), which target similar deployment goals.

**Questions:**

1. Authors assert “minor storage,” set Top-10 in all configs, but provide no concrete footprint/bandwidth numbers. To better understand the practicality of the Top-10–logits setup, could you share results, such as the approximate on-disk size per token when storing Top-10 logits and the total footprint for 0.5B tokens under typical context lengths?

2. Table 3 shows the ablation of the searching budget. Any clue why adding the search budget would even decrease performance?

3. How does E³-PRUNER compare with other LLM depth pruners? For example, LLM-Streamline[1], which identifies less important blocks and substitutes them with a light-weight block obtained via KD.

[1]Streamlining Redundant Layers to Compress Large Language Models, iCLR'25

---

> ### Author Response · Authors · 2025-11-21
>
> > W1: The behavioral consistency metric is weakly defined. The paper measures consistency as average accuracy on a small mixed set rather than teacher–student agreement (e.g., output match rate, output distributions, or log-prob correlations). This undermines the claim that KD better preserves behavior.
>
> **A1: The defined consistency score directly quantifies the alignment of the compressed model and the original model across diverse domain tasks.** We agree that metrics such as output match rate, output distributions, and log-probability correlations serve as useful intermediate indicators—indeed, knowledge distillation (KD) explicitly minimizes such discrepancies by aligning soft targets, which partly explains KD’s stronger consistency. However, these metrics remain relatively indirect proxies. In our work, we prioritize evaluating consistency from the perspective of end-task performance, which better reflects practical model behavior for real-world users. This definition directly captures what matters most to practitioners: whether the compressed model behaves like the original in actual applications. For example, task-specific results for Figure 6(b) are provided below and will be included in the revised manuscript to support this claim.
> | Iterations | |SFT                           || ||| KD                             |||
> |------------|------|--------|--------|----------|----|--------|--------|--------|----------|
> |            | MMLU   | GSM8K  | IFeval | Humaneval|| MMLU   | GSM8K  | IFeval | Humaneval|
> | 150        | 71     | 64     | 41     | 65       || 72     | 64     | 57     | 59       |
> | 300        | 73     | 61     | 32     | 13       || 73     | 76     | 53     | 64       |
> | 450        | 72     | 66     | 31     | 32       || 73     | 68     | 58     | 73       |
> | 600        | 73     | 68     | 26     | 31       || 72     | 78     | 53     | 74       |
> | 750        | 72     | 72     | 35     | 46       || 74     | 70     | 55     | 68       |
> | 900        | 73     | 62     | 34     | 46       || 74     | 75     | 57     | 70       |
> | 1050       | 76     | 60     | 31     | 48      | | 75     | 77     | 57     | 70       |
> | 1200       | 74     | 59     | 31     | 47       || 75     | 78     | 54     | 72       |

---

> ### Author Response · Authors · 2025-11-21
>
> > W2: Storage/IO for offline KD is unquantified. The method relies on offline Top-K logits and asserts “minor storage,” using Top-10 in all configs, but provides no concrete footprint or bandwidth numbers under long contexts and 0.5B-token training budgets. This limits reproducibility and deployability assessments.
>
> > Q1: Authors assert “minor storage,” set Top-10 in all configs, but provide no concrete footprint/bandwidth numbers. To better understand the practicality of the Top-10–logits setup, could you share results, such as the approximate on-disk size per token when storing Top-10 logits and the total footprint for 0.5B tokens under typical context lengths?
>
> **A2: During training, offline knowledge distillation (KD) incurs minimal GPU memory overhead—it primarily consumes inexpensive disk storage while placing negligible demand on precious GPU memory.** Specifically, to store the Top-10 logits (indices and values) for each token on disk, storage requirements increase by approximately 20×. For a dataset of 0.5B tokens, the original token_ids (stored as int32) occupy ~2 GB; storing the Top-10 logits thus requires an additional ~40 GB of disk space.
> However, during training, logits are loaded on-demand per training sequence. In our experiments with Qwen2.5-14B-Instruct, with a sequence length of 16K and micro-batch size of 2, the logits loaded into GPU memory per forward pass amount to only 16384×2×20×4 bytes≈2.6 MB. This represents a mere 2.6 MB of additional GPU memory compared to standard supervised fine-tuning (SFT), confirming that the runtime overhead of offline KD is negligible.
> Thanks for raising this point, and we will incorporate the above analysis into the revised manuscript to strengthen the argument.

---

> ### Author Response · Authors · 2025-11-21
>
> > W3: Fairness of the comparison. ShortGPT is training-free, so its lower accuracy vs. trained methods is expected; using it as a headline accuracy comparator can be misleading. A fair alternative is ShortGPT with the same recovery budget (SFT/KD) to normalize token use.
>
> **A3: We include ShortGPT primarily to underscore that, despite its training-free efficiency and minimal calibration requirements, it suffers from severe accuracy degradation—rendering it unsuitable for practical deployment (Figure 2).** In contrast, our method achieves a superior trade-off across the three key challenge dimensions and consistently outperforms other training-based approaches.
> In Table 1, we compare against the **Minitron strategy—which uses the same BI-based layer importance as ShortGPT and an identical distillation token budget**—and show clear gains: 58.3% vs. 55.3% on LLaMA-2-7B (60% pruning) and 61.9% vs. 60.1% on Qwen2.5-14B-Instruct (50% pruning), as also extracted to the table below.
> |||SciQ|PIQA|WG|ArcE|ArcC\(25\)|HS\(10\)|LogiQA|BoolQ|MMLU\(5\)|AVG|
> |---|---|---|---|---|---|---|---|---|---|---|---|
> |LLaMA\-2\-7B, 60% pruned |Minitron|86.4|67.2|56.5|58.0|32.9|49.4|29.3|62.4|\- |55.3|
> ||E$^3$\-Pruner|88.4  |71.2 |57.3 |65.4 |38.1 |54.6 |28.1 |63.6|\- |58.3|
> |Qwen2.5\-14B\-Instruct, 50% pruned|Minitron|91.9 |73.5 |62.0 |72.1 |44.9 |64.5 |28.4 |70.4 |32.8 |60.1|
> ||E$^3$\-Pruner|93.7    |76.9|63.0|76.0|47.9 |67.2|30.0 |66.5 | 36.5 |61.9|
>
> We also benchmark against ShearedLLaMA and DarwinLM under comparable training conditions, with our method again achieving state-of-the-art performance.

---

> ### Author Response · Authors · 2025-11-21
>
> > W4: Missing comparisons to recent depth-pruning baselines. The paper does not compare to other layer/depth pruners that replace blocks via KD (e.g., LLM-Streamline), which target similar deployment goals.
>
> > Q3: How does E³-PRUNER compare with other LLM depth pruners? For example, LLM-Streamline[1], which identifies less important blocks and substitutes them with a light-weight block obtained via KD.
>
> **A4: Our method significantly outperforms LLM-Streamline under a fair comparison.** LLM-Streamline replaces contiguous layers with a lightweight module and fine-tunes it for performance recovery. For a fair comparison, we adopt its layer replacement strategy but use the same distillation budget as in our method. On Qwen2.5-14B-Instruct with 50% of layers pruned, our E³-Pruner achieves 61.9% average accuracy versus 54.2% for LLM-Streamline.
> This improvement stems from our differentiable Gumbel-TopK search, which learns a globally optimized layer mask, whereas LLM-Streamline's fixed removal of contiguous layers imposes rigid structural constraints that hinder performance.
> ||SciQ|PIQA|WG|ArcE|ArcC\(25\)|HS\(10\)|LogiQA|BoolQ|MMLU\(5\)|AVG|
> |---|---|---|---|---|---|---|---|---|---|---|
> |LLM\-Streamline + Adaptive KD|89.2|73.5|52.6|68.7|36.0|55.9|25.5|59.9|26.9|54.2|
> |E$^3$\-Pruner|93.7    |76.9|63.0|76.0|47.9 |67.2|30.0 |66.5 |36.5 |61.9|

---

> ### Author Response · Authors · 2025-11-21
>
> > Q2: Table 3 shows the ablation of the searching budget. Any clue why adding the search budget would even decrease performance?
>
> **A5: Since we maintain a fixed total budget, increasing the searching-stage budget necessarily reduces the budget available for the fine-tuning stage.** Our experiments show that allocating 10% of the total budget to the searching-stage achieves the best overall trade-off.

---

### Author Response · Authors · 2025-11-30
**Rebuttal Summary**

We would like to provide a brief summary of our rebuttal and key clarifications below:

**1. Core Contributions & Impact**

We are pleased that reviewers have recognized our core contributions, which we summarize below:

  * **Problem Significance & Technical Innovation:** As noted by reviewers *gvMZ* and *6ac5*, our E³-Pruner address three practical challenges—task effectiveness, training economy, and inference efficiency—under real-world deployment constraints. We proposed a novel Gumbel-TopK based mask optimization and an entropy-aware adaptive knowledge distillation to jointly resolve these key trade-offs.
  * **Comprehensive Evaluation:** As unanimously acknowledged by all reviewers, our extensive experiments across diverse models and benchmarks have robustly demonstrated the effectiveness and practicality of E³-Pruner.
  * **Generalization and Scalability:** Our E³-Pruner generalizes across architectures and, to the best of our knowledge, is the first to successfully apply structured layer pruning to large MoE models (e.g., DeepSeek-R1), which is also recognized by reviewers *iaGe* and *Kz3N*.

**2. Rebuttal Highlights**

During the rebuttal, we received valuable suggestions and concerns from all reviewers. In response, we have provided detailed clarifications and additional experimental results. We conclude some common concerns below:

  * **Baseline Selection:** Beyond fine-tuning-based methods in the original manuscript, we have added comparisons with random pruning and recent depth-pruning methods (Shortened LLaMA, SLEB, LLM-Streamline), which further demonstrate the superiority of E³-Pruner over state-of-the-art baselines.
  * **Comparison Fairness:** We provide further clarifications of our experimental and evaluation settings to show that a fair comparison across methods has been rigorously ensured. Additionally, we present results for both baseline methods and E³-Pruner before and after the identical distillation. These results consistently show that E³-Pruner outperforms all baselines, further highlighting the effectiveness of our Gumbel-TopK mask search and adaptive knowledge distillation.
 * **Additional Ablations:** We have added new ablation studies on the iterative pruning–distillation scheme and the annealing schedule, directly addressing concerns raised from reviewers *Gw2L* and *6ac5*. These analyses not only validate the robustness and generalization of our E³-Pruner but also reinforce the reliability of our method under different training configurations and hyperparameter settings.

We sincerely appreciate all reviewers for their insightful feedback, which has significantly strengthened our work. We believe that our rebuttal has comprehensively addressed all raised concerns.

---

### Meta-Review · Area_Chair_5prQ · 2026-01-07

**Summary:**

The paper proposes E³-Pruner, a layer pruning framework for LLMs that combines a differentiable Gumbel-TopK mask sampler with an entropy-aware adaptive knowledge distillation strategy. While the authors demonstrated the method's application across various models including DeepSeek-R1, significant concerns remain regarding the novelty of the technical components and the marginal utility of the search phase compared to simple heuristics. Critical issues persist regarding the necessity of the complex search mechanism given that random baselines perform comparably, and the lack of distinct methodological innovation beyond combining existing techniques. Consequently, the paper is not recommended for acceptance.

**Reviewer Concerns:**

**Addressed Concerns:**

1.  **Baseline Comparisons** (Reviewer gvMZ, Reviewer Gw2L, Reviewer Kz3N): The authors addressed requests for broader comparisons by adding results for Shortened LLaMA, SLEB, and LLM-Streamline, and clarifying configurations for Minitron.
2.  **Implementation Details** (Reviewer iaGe): The authors clarified the fine-tuning setup for DeepSeek-R1 (LoRA with 64 NPUs) in the rebuttal.

**Outstanding Concerns:**

1.  **Marginal Utility of the Search Strategy** (Reviewer Kz3N, Reviewer Gw2L): Reviewer Kz3N highlighted a critical issue where a random pruning strategy achieved performance very close to the proposed method (59.9% vs. 61.9%). This small margin undermines the central premise that a complex, differentiable Gumbel-TopK search is necessary or "Economical," as random selection requires zero search cost. Reviewer Gw2L similarly noted that heuristic methods like SLEB combined with the authors' distillation achieved 60.9%, marginally different from E³-Pruner's 61.9%.
2.  **Limited Novelty** (Reviewer Gw2L, Reviewer 6ac5): Reviewer Gw2L and Reviewer 6ac5 argued that the core technical contributions are straightforward adaptations of existing methods (Gumbel-Softmax/TopK) and weighting schemes, offering negligible conceptual advancement. The rebuttal did not effectively counter the argument that this is primarily an application of known techniques rather than a methodological breakthrough.
3.  **Source of Gains** (Reviewer Gw2L): The ablation studies suggest that the majority of the performance gain comes from the Adaptive Knowledge Distillation (KD) rather than the pruning mask search itself. Since the pruning search yields results comparable to random or heuristic selection (SLEB), the technical contribution of the "E³" framework as a pruning selector is questionable.

**Reviewer Scores:**

1.  **Reviewer Gw2L:** Score likely remains unchanged (2). The reviewer explicitly stated that the additional experiments did not fully demonstrate the effectiveness of the method over random/heuristic baselines.
2.  **Reviewer Kz3N:** Score likely remains unchanged (4). The reviewer remained skeptical after the random baseline comparison showed minimal gaps.
3.  **Reviewer 6ac5:** Score likely remains unchanged (4). The concerns regarding novelty and straightforward heuristics were not resolved.
4.  **Reviewer gvMZ:** Score likely remains unchanged (6). While satisfied with some clarifications, the fundamental issues raised by other reviewers regarding the search utility weigh heavily.

---

### Decision · Program_Chairs · 2026-01-26

Reject